# Self-powered H$_2$ production with bifunctional hydrazine as sole consumable

Xijun Liu[1], Jia He[1], Shunzheng Zhao[2], Yunpeng Liu[3], Zhe Zhao[1], Jun Luo [1], Guangzhi Hu[4], Xiaoming Sun[5] & Yi Ding [1]

Splitting hydrazine into H$_2$ and N$_2$ by electro-catalyzing hydrogen evolution and hydrazine oxidation reactions is promising for replacing fossil energy with H$_2$. However, current hydrazine splitting is achieved using external powers to drive the two reactions, which is inapplicable to outdoor use. Here, Fe-doped CoS$_2$ nanosheets are developed as a bifunctional electrocatalyst for the two reactions, by which direct hydrazine fuel cells and overall-hydrazine-splitting units are realized and integrated to form a self-powered H$_2$ production system. Without external powers, this system employs hydrazine bifunctionally as the fuel of direct hydrazine fuel cell and the splitting target, namely a sole consumable, and exhibits an H$_2$ evolution rate of 9.95 mmol h$^{-1}$, a 98% Faradaic efficiency and a 20-h stability, all comparable to the best reported for self-powered water splitting. These performances are due to that Fe doping decreases the free-energy changes of H adsorption and adsorbed NH$_2$NH$_2$ dehydrogenation on CoS$_2$.

[1] Center for Electron Microscopy and Tianjin Key Lab of Advanced Functional Porous Materials, Institute for New Energy Materials & Low-Carbon Technologies, School of Materials, Tianjin University of Technology, Tianjin 300384, China. [2] Department of Environmental Engineering, University of Science and Technology Beijing, Beijing 100083, China. [3] Institute of High Energy Physics, Chinese Academy of Sciences, Beijing 100049, China. [4] Key Lab for Chemistry of Plant Resources in Arid Regions, State Key Lab for Basis of Xinjiang Indigenous Medicinal Plants Resource Utilization, Xinjiang Technical Institute of Physics and Chemistry, Chinese Academy of Sciences, Urumqi 830011, China. [5] State Key Lab of Chemical Resource Engineering, Beijing University of Chemical Technology, Beijing 100029, China. These authors contributed equally: Xijun Liu, Jia He. Correspondence and requests for materials should be addressed to J.L. (email: jluo@tjut.edu.cn) or to G.H. (email: gzhu2004@126.com) or to Y.D. (email: yding@tjut.edu.cn)

Currently increasing petroleum consumption and global pollution have triggered an urgent demand to find clean energy sources with renewability, for which hydrogen ($H_2$) is considered as a promising candidate due to its zero $CO_2$ emission and high energy density[1–8]. Thus, electrocatalytic water and hydrazine splittings have been developed as two green approaches to produce $H_2$ (refs. [9–17]), each of which consists of two reactions, hydrogen evolution reaction (HER) and water oxidation reaction (WOR) for water splitting and HER and hydrazine oxidation reaction (HzOR) for hydrazine splitting. HzOR has a lower overpotential to overcome than WOR[16,17]. Moreover, when no separation methods are used, the product of hydrazine splitting is the mix of $H_2$ and $N_2$, much safer than that of water splitting ($H_2$ and $O_2$)[9–17]. However, a major hindrance of current hydrazine splitting is that an external power supply is required for driving its electrochemical process[16,17], causing it not to be applicable to mobile devices, vehicles and field activities. Hence, it is greatly desired to develop a self-powered electrocatalytic $H_2$ generation system.

Some pioneering breakthroughs[10–12,18–20] have been reported since the self-powered concept without external power supplies was proposed by Brown and Sheen in 1962 (ref. [21]). For example, various self-powered electronic or electrochemical applications, such as pollution cleanup, electrooxidation, particulate filtering, and $H_2$ production, have been realized with triboelectric/piezo-electric nanogenerators or batteries[10–12,18–20]. Among them, five self-powered systems integrating electrocatalytic overall-water-splitting (OWS) units and Zn–air batteries/nanogenerators/thermoelectric cells/solar cells have been developed with great success, and they can produce $H_2$ using water and Zn or mechanical/thermal/solar energies as consumables[10–12,18,19]. However, no self-powered $H_2$ production systems with hydrazine splitting have been reported so far.

Inspired by the above breakthroughs and on the basis of our previous works about HER and fuel cells[3,22–24], we develop herein an efficient and stable bifunctional catalyst consisting of thin Fe-doped $CoS_2$ (Fe-$CoS_2$) nanosheets for HER and HzOR. Their performances are comparable to the best results published in the literature: for HER, the Fe-$CoS_2$ nanosheets exhibit a Pt-like activity with low overpotential ($\eta$) values at 10 mA cm$^{-2}$ (40 mV in 1.0 M KOH, 31 mV in 0.5 M $H_2SO_4$, and 49 mV in 1.0 M phosphate buffer solution (PBS)), high stabilities at all-pH values and 93–98% Faradaic yields; for HzOR, the required working potential for obtaining the current density ($j$) of 100 mA cm$^{-2}$ on the Fe-$CoS_2$ nanosheets in 1.0 M KOH is only 129 mV, which is comparable to that of Pt/C (170 mV). Thus, we use the Fe-$CoS_2$ nanosheets as anodes to assemble direct hydrazine fuel cells (DHzFCs) with $H_2O_2$ or $O_2$ as oxidizing agents, which have been recognized as a green type of energy devices without $CO_2$ emission[23,24]. Their maximum power density ($P_{max}$) values are found to be 246 mW cm$^{-2}$ ($H_2O_2$) and 125 mW cm$^{-2}$ ($O_2$), which are both among the best values published for DHzFCs with Co-based electrocatalysts. We also use the Fe-$CoS_2$ nanosheets as anodes and cathodes to perform overall hydrazine splitting (OHzS), and the OHzS operating voltage is found to be as low as 0.95 V for achieving 500 mA cm$^{-2}$, also among the best results of all reported OHzS works. Further, we integrate a DHzFC and an OHzS unit to fabricate a system for self-powered $H_2$ production, in which hydrazine is the sole consumable serving bifunctionally as the DHzFC fuel and the splitting target. The DHzFC anode and the two electrodes of the OHzS unit are all the Fe-$CoS_2$ nanosheets. This self-powered system exhibits a good stability at 0.7 V for 20 h and a hydrogen evolution rate of 9.95 mmol h$^{-1}$, comparable to the best reported for self-powered $H_2$ production system based on water splitting[10–12,18,19]. Theoretical calculations show that the above efficient performances are caused by the Fe

doping, which can decrease the free-energy changes of not only the H adsorption but also the dehydrogenation of adsorbed $NH_2NH_2$ (denoted as $NH_2NH_2^*$) on $CoS_2$. These results demonstrate that the DHzFC-driven OHzS is a promising strategy for real-time hydrogen generation without any external power supplies.

## Results

**Synthesis and characterization of the Fe-$CoS_2$ nanosheets**. The synthesis procedure for the Fe-$CoS_2$ nanosheets is presented in Methods and Supplementary Fig. 1: Firstly, layered Fe-doped Co $(OH)_2$ precipitates, which were intercalated with dodecyl sulfate ions, were synthesized by performing hexamethylenetetramine hydrolysis in an aqueous solution containing $Co^{2+}$, $Fe^{2+}$, and sodium dodecyl sulfate[8]. After this, the obtained precipitates were mixed with formamide, realizing the exfoliation of their host layers, and the resultant suspension was ultrasonicated for 12 h. Then, the suspension was centrifuged, in which powders were collected and found to be thin hydroxide nanosheets (Supplementary Figs. 2 and 3). Finally, the hydroxide nanosheets were calcined in the presence of S powder, leading to the formation of the Fe-$CoS_2$ nanosheets.

Figure 1a gives an X-ray diffraction (XRD) pattern of the Fe-$CoS_2$ nanosheets. The diffraction peaks can all be indexed to standard cubic $CoS_2$, whose powder diffraction file (PDF) number is 89-1492, and neither impurity peaks nor peak shifts exist, indicating that the content of Fe dopants is so low that the $CoS_2$ crystal lattice of the nanosheets is not disturbed (see Supplementary Figs. 3 and 4 for more details). This result is consistent with Raman measurement (Supplementary Fig. 5). Transmission electron microscopy (TEM) imaging (Fig. 1b) illustrates that the nanosheets are so thin that they are nearly transparent to the electron beam[25]. Their thickness was measured by atomic force microscopy (AFM) to be within a narrow range of 1.22 ± 0.03 nm (Fig. 1c, d and Supplementary Fig. 6). Further, high-angle annular dark-field (HAADF) imaging with the corresponding electron energy-loss spectroscopy (EELS) mapping of elements (Fig. 1e) indicates that the distribution of Fe, Co, and S in the nanosheets is homogeneous. High-resolution TEM (HRTEM) imaging confirms that the nanosheets have the crystal lattice with the same spacing values as those of the standard $CoS_2$ phase, such as the values of 0.247 ± 0.001 and 0.278 ± 0.002 nm for the {012} and {200} crystal planes in Fig. 1f, which equal the theoretical values (0.2477 and 0.2769 nm) of {012} and {200}, respectively, from $CoS_2$ PDF 89-1492 within the error bars. Brunauer–Emmett–Teller (BET) measurements were conducted, and their results indicate that the Fe-$CoS_2$ nanosheets and pure $CoS_2$ ones possess similar specific surface areas and pore properties (Supplementary Fig. 7), also demonstrating that the Fe doping did not change the morphology and structure of the nanosheets. In addition, the atomic percentage of Fe dopants in the nanosheets was measured using inductively coupled plasma mass spectrometry (ICP-MS) to be 5.1 at.% (the Fe atomic percentage can be tuned by adjusting the $Fe^{2+}$ loading in the synthesis, and we found the 5.1 at.% value to correspond to the best HER performance, whose details will be shown later).

**Electrocatalytic activity and durability toward HER and HzOR**. To characterize the HER electrocatalytic performance, the Fe-$CoS_2$ nanosheet catalyst was deposited on an electrode of glassy carbon with a mass loading of 20 μg cm$^{-2}$ and was then used as the working electrode in a typical three-electrode cell, which contained 1.0 M KOH and used a graphite plate as the counter electrode (see Methods for more details). The correction of ohmic potential drop loss due to solution resistance was performed to all

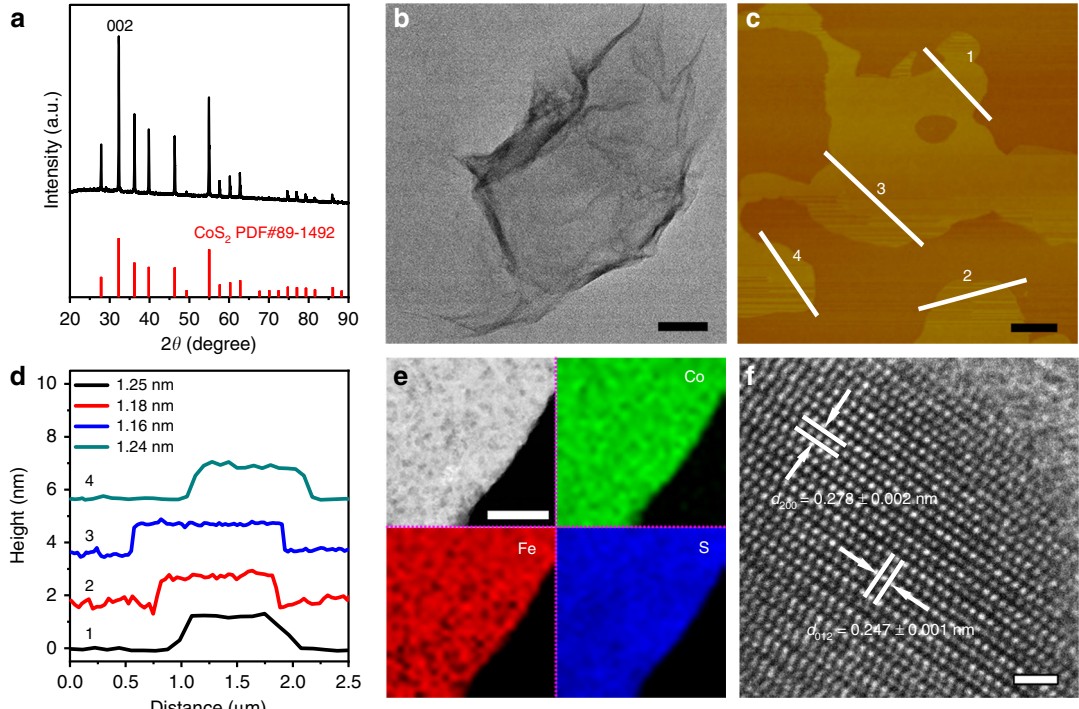

**Fig. 1** Characterization of the Fe-CoS$_2$ nanosheets. **a** XRD pattern, in which the standard pattern of CoS$_2$ is added in red for comparison. **b** Low-magnification TEM image. **c** AFM image. **d** Height profiles taken along the white lines of 1–4 in **c**. The thickness values measured by the profiles are labeled in **d**. **e** HAADF image and its corresponding EELS mapping results of Co, Fe, and S. **f** HRTEM image. Scale bars: **b**, 100 nm; **c**, 500 nm; **e**, 100 nm; **f**, 1 nm

initial data[3], and all potential values shown for HER in this study were relative to the reversible hydrogen electrode (RHE). By the electrochemical measurements, it is found (Supplementary Fig. 8) that when the atomic percentage of Fe was changed from 0 to 9.5 at.%, the HER activity of the catalyst increased firstly, reached the top value at 5.1 at.% and then decreased (the changing reason will be shown later). Thus, the 5.1 at.% sample was used for all the following experiments. In addition, commercial Pt/C (20 wt.%, Johnson Matthey) was measured as a control sample. Besides, the effect of Fe-CoS$_2$ loading on the HER activity was also examined (Supplementary Fig. 9a, b), showing that when the loading increased, the overpotential for the HER current density of 10 mA cm$^{-2}$ (denoted as $\eta_{10}$) initially decreased, reached the lowest point at 20 μg cm$^{-2}$ and then increased (this increase of $\eta_{10}$ after 20 μg cm$^{-2}$ was possibly due to the slow electron-transfer kinetics caused by the over-loading of catalysts[26]). This change indicates that the HER performance of Fe-CoS$_2$ was the highest at 20 μg cm$^{-2}$. Thus, we used this loading for the HER experiments.

In Fig. 2a, the 5.1 at.% Fe-CoS$_2$ nanosheets show an obviously enhanced HER activity, such as $\eta_{10} = 40$ mV, compared to pure CoS$_2$ ($\eta_{10} = 204$ mV) in 1.0 M KOH. Their HER activity is also better than those of other reported CoS$_2$-based electrodes and among the best results of precious-metal-free HER electrocatalysts reported under comparable conditions (Supplementary Table 1). More importantly, compared to Pt/C, the Fe-CoS$_2$ nanosheets exhibit larger $\eta$ values only when $j < 20$ mA cm$^{-2}$, and their $\eta$ values for $j > 20$ mA cm$^{-2}$ are much smaller, indicating better HER activities than Pt/C at-large $j$. This HER behavior of Fe-CoS$_2$ can be explained as follows: Pt/C exhibited a typical 0 mV overpotential at the onset position, and its corresponding HER current density ($j_{HER}$) started to increase from 0 earlier and faster than that of Fe-CoS$_2$; but, when the applied potential was increased further, the increase rate of $j_{HER}$ of Pt/C was smaller than that of Fe-CoS$_2$, due to the larger charge transfer resistance of Pt/C (see Supplementary Fig. 10 for details); thus, after the

applied potential was increased to be greater than a critical value (that is, $j_{HER}$ was greater than a critical value), $j_{HER}$ of Fe-CoS$_2$ started to be larger than that of Pt/C, as displayed in Fig. 2a. Similar cases have been reported on other electrocatalysts, such as MoS$_2$ (ref. [27]) and Ni-Mo alloy[28]. The Tafel slopes for Pt/C, CoS$_2$ and Fe-CoS$_2$ were obtained to be 35, 87, and 32 mV dec$^{-1}$, respectively (Fig. 2b), suggesting that the HER over Fe-CoS$_2$ conforms to the known Volmer–Heyrovsky mechanism and that the rate-limiting step is the electrochemical desorption step[3,4]. To assess electrochemical active surface area (ECSA) for the HER experiments, double layer capacitance ($C_{dl}$) values were measured for achieving the ECSA values[3,9,16] (see Methods and Supplementary Fig. 11a–c for details). Consequently, Fe-CoS$_2$ was found to have a larger ECSA value (900 cm$^2$) than CoS$_2$ (683 cm$^2$), indicating the presence of more active surface areas on Fe-CoS$_2$ than on CoS$_2$. Figure 2c displays that Fe-CoS$_2$ kept a nearly constant value of current density at $\eta = 200$ mV for 40 h and had a nearly unchanged HER electrocatalytic performance after 10,000 potential cycles. Further, the electrochemical impedance spectroscopy (EIS) results of Fe-CoS$_2$ before and after the stability test in Fig. 2c were found to be almost identical to each other (Supplementary Fig. 12), indicating that Fe-CoS$_2$ has stable electrocatalytic kinetics through the stability test. The above findings agree well with the structural stability of Fe-CoS$_2$ through the chronoamperometric test (see details in Supplementary Fig. 13a, b).

Likewise, the HER activities of Fe-CoS$_2$ in 0.5 M H$_2$SO$_4$ and 1.0 M PBS were also measured and found to be among the best values of HER electrocatalysts without precious metals reported under comparable conditions (Supplementary Fig. 14 and Table 1). For instance, the $\eta_{10}$ value for Fe-CoS$_2$ in 0.5 M H$_2$SO$_4$ and 1.0 M PBS are 31 and 49 mV, respectively. The counterpart values for CoS$_2$ are 181 and 299 mV, while those for Pt/C are 11 and 44 mV. Moreover, Fe-CoS$_2$ exhibited much lower $\eta$ values than Pt/C at $j > 95$ mA cm$^{-2}$ in PBS. This catalyst also

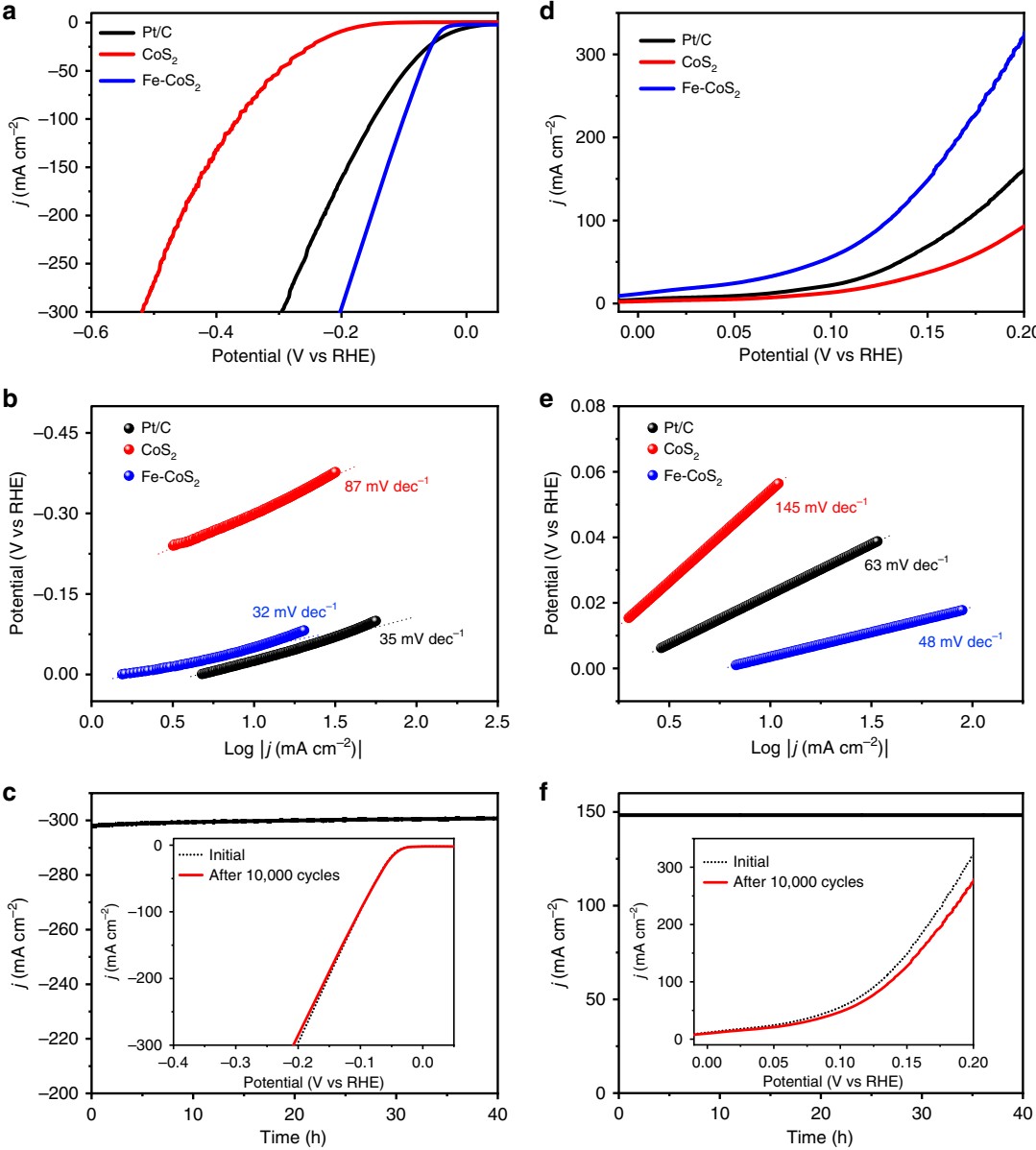

**Fig. 2** Electrochemical performances of the Fe-CoS$_2$ nanosheets for HER and HzOR in three-electrode cells containing alkaline electrolytes at room temperature. **a** HER polarization curves of 20 wt.% Pt/C, the pristine CoS$_2$ nanosheets and the Fe-CoS$_2$ ones. **b** Tafel plots from **a**. **c** Chronoamperometric curve recorded on the Fe-CoS$_2$ nanosheets for 40 h at a constant working potential of −0.2 V for HER, in which the onset value of current density is 298 mA cm$^{-2}$, very close to the one (296 mA cm$^{-2}$) at −0.2 V in **a**. The meanings of **d**–**f** are the same as those of **a**–**c**, but they are for HzOR. The used working potential in **f** is 0.15 V. The insets in **c**, **f** are the HER and HzOR polarization curves of the Fe-CoS$_2$ nanosheets before and after the nanosheets experienced 10,000 potential cycles, respectively. The electrolytes for HER and HzOR were 1.0 M KOH and 1.0 M KOH with 0.1 M hydrazine, respectively (see Methods for more details)

has stable electrocatalysis in H$_2$SO$_4$ and PBS. Further, the amounts of evolved H$_2$ gases were measured using gas chromatography, which give the Faradaic efficiencies (FEs) of Fe-CoS$_2$ to be 98%, 96 and 93% in 1.0 M KOH, 0.5 M H$_2$SO$_4$, and 1.0 M PBS, respectively (see details in Methods and Supplementary Fig. 15). These values suggest that the cathode currents mainly result from the generation of H$_2$.

It is well known that $\eta$, Tafel slope, and FE can be affected by electrocatalyst loading[9]. Hence, to exclude the influence of loading, it is important to evaluate intrinsic HER activities using turnover frequency (TOF), namely the rate of evolved H$_2$ molecules per active site[6,29,30]. The TOF values of the Fe-CoS$_2$ nanosheets were measured and calculated to be 6.74 s$^{-1}$ (1.0 M KOH), 4.76 s$^{-1}$ (0.5 M H$_2$SO$_4$), and 2.21 s$^{-1}$ (1.0 M PBS) at

$\eta = 100$ mV, also among the best values reported (see details in Methods, Supplementary Figs. 16, 17 and Table 2). In addition, when $\eta$ was 100 mV, the TOF values of Pt/C were 1.53, 5.46, and 2.05 s$^{-1}$ in 1.0 M KOH, 0.5 M H$_2$SO$_4$ and 1.0 M PBS, respectively (see Supplementary Fig. 16 for details). Comparing the two sets of TOF values indicates that the intrinsic HER activities of Fe-CoS$_2$ are higher than those of Pt/C in 1.0 M KOH and 1.0 M PBS but worse than that of Pt/C in 0.5 M H$_2$SO$_4$, in agreement with the HER measurement results (Fig. 2a and Supplementary Fig. 14a, b). Moreover, by the data in Fig. 2a, the HER exchange current density ($j_0$) values of CoS$_2$ and Fe-CoS$_2$ were obtained to be 0.01 and 0.89 mA cm$^{-2}$, respectively, revealing that Fe-CoS$_2$ has an intrinsic electrocatalytic activity superior to CoS$_2$ (the superiority reason will be discussed later), because larger $j_0$ values reflect

higher intrinsic electron-transfer rates between catalysts and electrolytes[2–8]. All of the above features imply that the Fe-CoS$_2$ nanosheets are promising for real-world hydrogen evolution applications over a wide pH-value range.

In addition to the efficient HER electrocatalytic performance, the Fe-CoS$_2$ nanosheets also efficiently catalyzed HzOR. The mass loading of Fe-CoS$_2$ in the HzOR experiments was kept at 20 μg cm$^{-2}$, because it is the optimum value for the HzOR (Supplementary Fig. 9c, d). As shown in Fig. 2d, the nanosheets delivered a $j_{HzOR}$ of 100 mA cm$^{-2}$ by a potential of 129 mV, namely $E_{100} = 129$ mV, and this value is comparable to that of Pt/C (170 mV) and smaller than that of CoS$_2$ (205 mV). The 129 mV value is also among the best reported for HzOR (Supplementary Table 3). Meantime, pure FeS$_2$ nanosheets were synthesized through a hydrolysis reaction[31] and a sulfurization process (see details in Methods and Supplementary Fig. 18). The FeS$_2$ nanosheet thickness was measured by AFM to be $1.31 \pm 0.04$ nm (Supplementary Fig. 18c, d), very close to that of the Fe-CoS$_2$ nanosheets ($1.22 \pm 0.03$ nm). In comparison to CoS$_2$ and Fe-CoS$_2$, the FeS$_2$ nanosheets exhibited an inferior HzOR activity with $E_{100} = 428$ mV (Supplementary Fig. 19). Moreover, the effect of the hydrazine concentration (from 0.1 to 1 M)[23,24] was measured on Fe-CoS$_2$ (Supplementary Fig. 20), showing that higher hydrazine concentrations correspond to larger HzOR current densities in the main range of applied potentials. This observation further confirms that the Fe-CoS$_2$ nanosheets are active for electro-catalyzing the HzOR. Figure 2e displays that the Tafel slope values of the Fe-CoS$_2$ nanosheets, the CoS$_2$ and Pt/C are 48, 145, and 63 mV dec$^{-1}$, respectively. Such a low value for the Tafel slope indicates the superior HzOR kinetics of the Fe-CoS$_2$ nanosheets. Meantime, Fe-CoS$_2$ was found to have a larger ECSA value (1200 cm$^2$) than CoS$_2$ (817 cm$^2$) for the HzOR, suggesting that more active surface areas exist on Fe-CoS$_2$ (Supplementary Fig. 11d–f). Further, the chronoamperometric response of the nanosheets (Fig. 2f) shows that no obvious drop of current was detected during 40 h, revealing their high stability for HzOR. This result is confirmed by EIS analysis (Supplementary Fig. 12b) and the corresponding TEM imaging and XRD patterns (Supplementary Fig. 13c, d). Moreover, the HzOR activity was retained well with a small increase of only 8 mV for $E_{100}$ after 10,000 potential cycles, as shown in the inset of Fig. 2f. In addition, it should be noted that CoS$_2$- and FeS$_2$-based electrocatalysts have received intensive attention because of their high performances[32–34], for which the Fe-CoS$_2$ nanosheets with the CoS$_2$ crystal structure unchanged is a new member.

**Performances of DHzFCs with Fe-CoS$_2$ as anodes**. Because of their high activity and durability toward HzOR, the Fe-CoS$_2$ nanosheets were used as an anode in a homemade DHzFC (Fig. 3a), by which N$_2$H$_4$ was oxidized to N$_2$ through HzOR[23,24]. The optimized mass loading of Fe-CoS$_2$ for DHzFCs was 1.5 mg cm$^{-2}$ (Supplementary Fig. 22). In the cathode made of Pt/C, H$_2$O$_2$ was reduced to H$_2$O (refs. [23,24]). We also assembled and measured another two DHzFCs with the CoS$_2$ nanosheets and Pt/C as anodes. Their cell performances are shown in Fig. 3b, giving their open-circuit voltage (OCV) values to be 1.80 V (Fe-CoS$_2$), 1.65 V (CoS$_2$) and 1.71 V (Pt/C). This comparison indicates the Pt-like intrinsic HzOR activity of the Fe-CoS$_2$ nanosheets[23,24]. Figure 3b also gives the $P_{max}$ value of the Fe-CoS$_2$ DHzFC to be 246 mW cm$^{-2}$, and this value is among the best performances reported so far (Supplementary Table 4).

Despite the above achievements, strongly oxidizing H$_2$O$_2$ molecules are not safe for the usage and transportation of DHzFCs. Thus, DHzFCs using O$_2$ were assembled and examined, as displayed in Fig. 3c, d. Their OCV values are 1.03 V (Fe-CoS$_2$),

0.95 V (CoS$_2$) and 0.98 V (Pt/C), also indicating the Pt-like intrinsic HzOR activity of the Fe-CoS$_2$ nanosheets. Their $P_{max}$ values are 125 mW cm$^{-2}$ (Fe-CoS$_2$), 36 mW cm$^{-2}$ (CoS$_2$) and 66 mW cm$^{-2}$ (Pt/C), of which the 125 mW cm$^{-2}$ value is close to the best reported (Supplementary Table 4). In addition, DHzFCs bubbled with air were also examined (Supplementary Fig. 23), and their $P_{max}$ values corresponding to Fe-CoS$_2$, CoS$_2$, and Pt/C were found to be 35, 13, and 21 mW cm$^{-2}$, respectively. These values of $P_{max}$ are much smaller than those of the O$_2$ DHzFCs. This is because air is not pure O$_2$. These results imply the promising potential of the Fe-CoS$_2$ nanosheets for developing safe and efficient DHzFCs.

**Electrochemical OHzS with and without external power sources**. Encouraged by the high activities and stabilities of the Fe-CoS$_2$ nanosheets toward HER and HzOR, we further investigated their application as an OHzS electrocatalyst, which was used bifunctionally as cathode and anode in a two-electrode electrolyzer (namely an OHzS unit, whose product includes H$_2$ and N$_2$; see Supplementary Movie 1 for its optical image). Experimentally, the optimized mass loading of Fe-CoS$_2$ for the OHzS unit was 0.5 mg cm$^{-2}$ (Supplementary Fig. 24), and this electrolyzer used a cell voltage of 0.95 V to achieve an OHzS current density of 500 mA cm$^{-2}$ in 1.0 M KOH and 0.1 M hydrazine(Fig. 4a). This cell voltage is much smaller than those of the CoS$_2$ and Pt/C counterparts (Fig. 4a) and also among the best values of previously reported bifunctional catalysts (Supplementary Table 5). Moreover, appreciable H$_2$ and N$_2$ bubbles from the anode and cathode surfaces were observed (Supplementary Movie 1). For comparison, an OWS electrolyzer was also tested using the Fe-CoS$_2$ nanosheets as both cathode and anode. It required a cell voltage of 3.30 V to afford 500 mA cm$^{-2}$ (Supplementary Fig. 25), suggesting a gargantuan energy consumption. Further, there was no obvious degradation found on the OHzS current density of the Fe-CoS$_2$ nanosheets during a 40-h continuous electrolysis (Fig. 4b), showing an excellent durability, which is consistent with the tinily changed OHzS performance of the nanosheets through 10,000 potential cycles (the inset of Fig. 4b).

The high OHzS performances shown in Fig. 4a, b were achieved by using an external power source, which was an electrochemical workstation (see details in Methods). But, it is generally difficult to find an external power source for outdoor applications, such as mobile devices, vehicles, and field activities[10–12,18,19]. Thus, we integrated an OHzS unit like that corresponding to Fig. 4a and a DHzFC like that in Fig. 3c, achieving a self-powered H$_2$ production system (Fig. 4c), in which hydrazine is the sole consumable serving bifunctionally as the DHzFC fuel and the splitting target. When this system was working, H$_2$ and N$_2$ evolved continuously at the OHzS cathode and anode, respectively (Supplementary Movie 2). The H$_2$ and N$_2$ yields were measured using gas chromatography, and the results are displayed in Fig. 4d, from which a linear relationship is shown between the H$_2$ yield and the elapsed time. This linear relationship gives the H$_2$ production rate to be 9.95 mmol h$^{-1}$ (see Supplementary Fig. 27 for more details), which is comparable to the best-reported values from self-powered H$_2$ production systems with water splitting, such as 0.08–8 mmol h$^{-1}$ (refs. [10–12,18,19]). Moreover, the working voltage of 0.7 V for this production rate was nearly unchanged for 20 h (Supplementary Fig. 28), suggesting that the system has a high stability for practical applications. In addition, Fig. 4d shows that the molar ratio of produced N$_2$ and H$_2$ is close to 1:2, conforming to the stoichiometry of hydrazine. Using the method in Supplementary Fig. 15, the FE of the system to produce H$_2$ was found to be 98%, close to 100%. Besides, it should be noted that in the individual

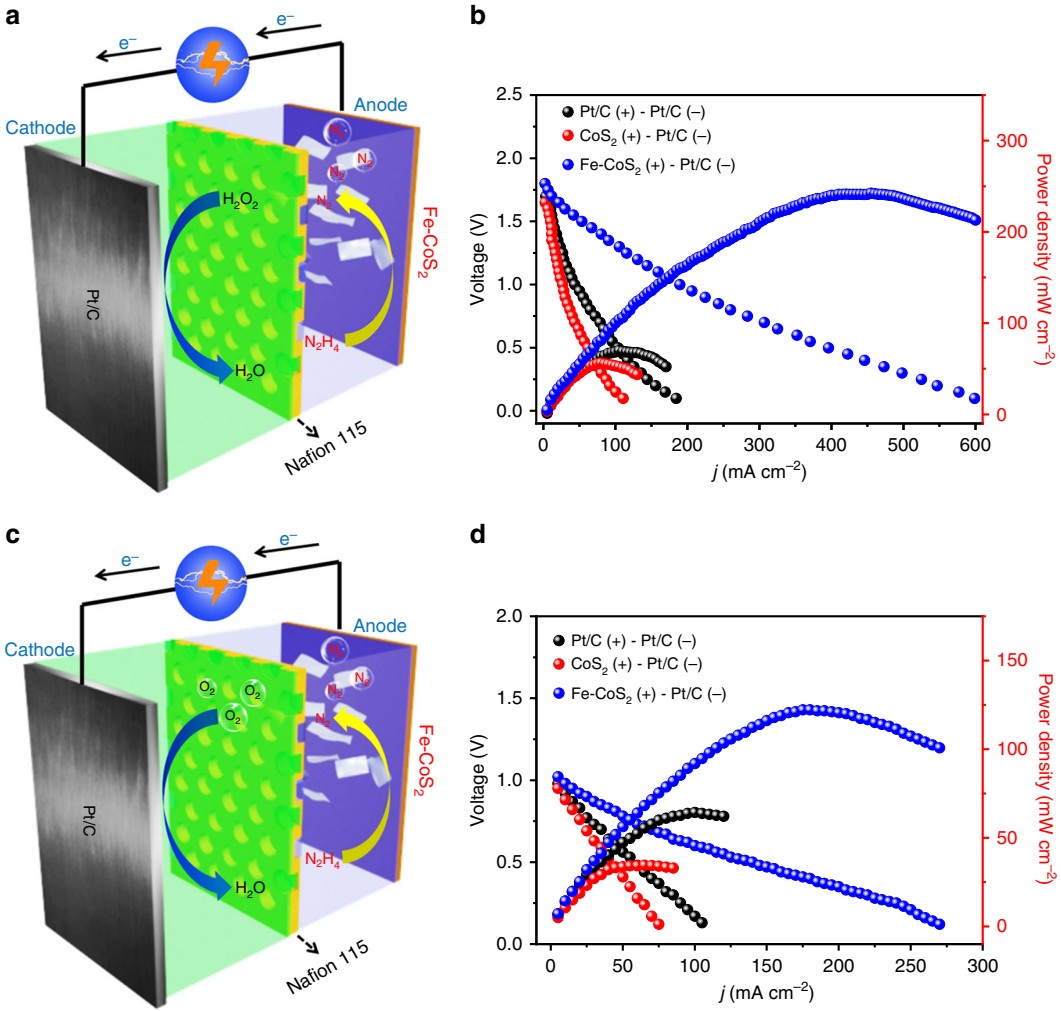

**Fig. 3** Application of the Fe-CoS$_2$ nanosheets in DHzFCs. **a** Schematic image of a DHzFC assembled with the Fe-CoS$_2$ nanosheets as the anode and commercial 40 wt.% Pt/C as the cathode (the oxidizing agent is H$_2$O$_2$). **b** Current density ($j$)–voltage ($V$) and current density ($j$)–power density ($P$) plots for the DHzFC in **a** and two other DHzFCs with the CoS$_2$ nanosheets and Pt/C as anodes. **c**, **d** have the same meanings as **a**, **b** but for DHzFCs with O$_2$ as the oxidizing agent. The optical images of the DHzFCs in **a**, **c** are in Supplementary Fig. 21. The mass loadings of the nanosheets and Pt/C in each corresponding electrode were 1.5 and 2 mg cm$^{-2}$, respectively. The cell temperature for each DHzFC was 80 °C, which has been widely used for DHzFC evaluation[23,24]

HER and HzOR measurements (Fig. 2a, d), the typical three-electrode system was used, and all of the potentials in the HER and HzOR polarization curves are values relative to the reference electrode. OHzS measurements are different, in which a two-electrode system was employed (Fig. 4a), and the voltages reported by the system reflect the real driving forces between the two electrodes, which are not the relative potentials. That is, potential and voltage are two different concepts here, and thus their values do not equal each other. Such cases have been reported on other electrocatalysts, such as Ni-Co/MoS$_2$ complexes[9] and Zn-doped Co$_3$O$_4$ sheets[15].

Additionally, the DHzFC in our self-powered H$_2$ production system with hydrazine as a sole consumable can be substituted by a methanol/ethanol fuel cell, a solar cell, an AA battery or other power source. But, if the substitution is done, not only hydrazine but also a second consumable (such as methanol, ethanol, sunlight, or AA battery) will be needed by the system, which will make the prospective application of the system less convenient. Meantime, the high toxicity and instability of hydrazine can be major hurdles to overcome for its wide-spread applications. Fortunately, a detoxification technique has been proposed to deal with the problems[35]: a hydrazine-fixing polymer can be synthesized by mixing methyl vinyl ketone and sodium 4-vinylbenzene sulfonate and cross-linking them with methylene(bis)acrylamide; the polymer contains carbonyl (>C=O) groups; the groups can react with hydrazine to form hydrazone (>C=N-NH$_2$) groups, which are harmless and stable on the polymer; when hydrazine is needed, hydrazine molecules can be released from the hydrazone groups by placing the polymer in water or a KOH aqueous solution. The polymer and KOH aqueous solution are both reusable, and so they are not consumables. Moreover, the polymer is harmless and stable, and KOH aqueous solutions containing hydrazine are already used in our self-powered system. Therefore, the detoxification technique should be an effective method to overcome the high toxicity and instability of hydrazine.

## Discussion

The efficient performance of the system originates from the high performances of the Fe-CoS$_2$ nanosheets for catalyzing HER and HzOR. Thus, density functional theory (DFT) calculations were used to explore the electrocatalysis mechanisms of HER and HzOR on the Fe-CoS$_2$ surface. Figure 5a–d shows the models of

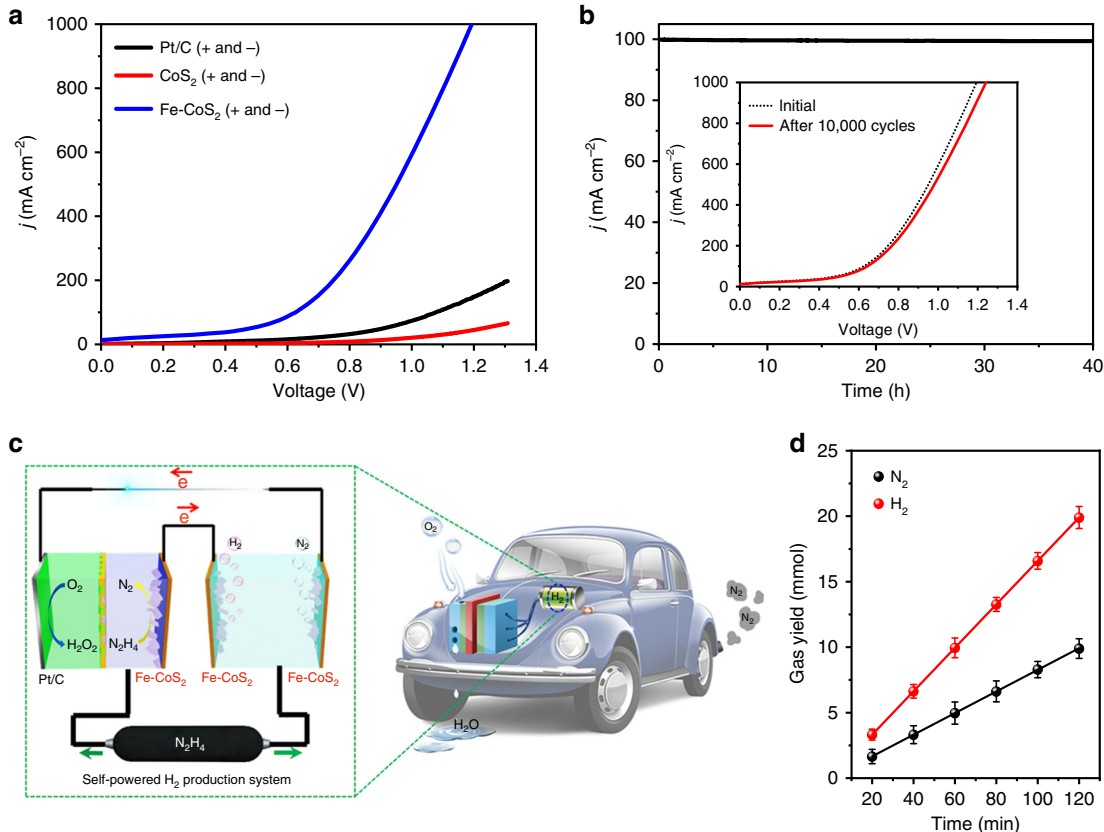

**Fig. 4** OHzS performances of the Fe-CoS$_2$ nanosheets with external power supplies and the realization of a self-powered H$_2$ production system with the nanosheets. **a** OHzS polarization curves of 20 wt.% Pt/C, the pure CoS$_2$ nanosheets and the Fe-CoS$_2$ ones, which all worked bifunctionally as anodes and cathodes with the same electrocatalyst loading of 0.5 mg cm$^{-2}$. **b** Chronoamperometric response recorded on the Fe-CoS$_2$ nanosheets for 40 h at a fixed working voltage of 0.63 V for OHzS. The inset in **b** is the OHzS polarization curves of the Fe-CoS$_2$ nanosheets before and after the nanosheets experienced 10,000 potential cycles. Each curve in **a**, **b** was measured in a two-electrode electrolyzer (namely an OHzS unit; see Supplementary Movie 1 for its optical image) containing 0.1 M hydrazine and 1.0 M KOH at 25 °C, and the electrolyzer was connected to an external power supply (an electrochemical workstation). **c** Schematic illustration of a self-powered H$_2$ production system integrating a DHzFC and an OHzS unit (see an optical image in Supplementary Fig. 26), in which hydrazine is the sole consumable serving bifunctionally as the DHzFC fuel and the splitting target. The DHzFC anode and the two OHzS electrodes are all the Fe-CoS$_2$ nanosheets. **d** Generated amounts of H$_2$ and N$_2$ in the system with the hydrazine concentration of 5.3 M at 0.7 V and room temperature. The 5.3 M concentration is an optimized value (see details in Supplementary Fig. 27). The error bars are made by the standard deviations of the amounts

CoS$_2$ (001) and Fe-CoS$_2$ (001) surfaces. The doping of Fe atoms was achieved using an Fe atom to replace one of the two surface Co atom in each unit cell at the CoS$_2$ surface, because the atomic diameter and chemical property of Fe are close to those of Co.

According to the above results, it is rational to speculate that if the Fe doping content in an Fe-CoS$_2$ sample is too small, the HER improvement effect should be very tiny. Thus, increasing the Fe doping content can cause this effect to be more obvious. But, if the content of Fe doping is too high, the crystal lattice of CoS$_2$ would be destroyed, making the effect worsen. The above speculation is in agreement with the experimental result in Supplementary Fig. 8, which shows that when the Fe content rose from 0 to 9.5 at.%, the HER activity of the Fe-CoS$_2$ nanosheets initially increased, reached the top at 5.1 at.% and then decreased. Moreover, XRD measurements show that the new crystal phases of CoS, Co$_9$S$_8$, and Fe$_3$S$_4$ appear in the 9.5 at.% sample (Supplementary Fig. 31)

An HER process can be simplified as that in Fig. 5e (refs. [4,6,36]), which shows three states including an initial pair of e− and H$^+$, an intermediate H$^*$ adsorbed on the CoS$_2$ or Fe-CoS$_2$ surface, and a final product of 1/2H$_2$. Previous DFT studies have indicated that in sulfides, metal atoms are active sites for HER, while S atoms are not[37]. Hence, our DFT work calculated the free-energy change of

the step between the first two states ($\Delta G_{H^*}$) for H$^*$ on the Co site of the CoS$_2$ surface and on the Co and the Fe sites of the Fe-CoS$_2$ surface, and the $\Delta G_{H^*}$ values were given to be +0.24 eV (the Co site of CoS$_2$), +0.15 eV (the Co site of Fe-CoS$_2$) and +0.35 eV (the Fe site of Fe-CoS$_2$), as displayed in Fig. 5e and Supplementary Fig. 29. The two figures also indicate: this step is the potential-determining step (PDS), because only its free-energy change is positive; for Fe-CoS$_2$, the H adsorption on the Fe site is less stable than on the Co, and not the Fe site but the Co one is the active site, because +0.15 eV < +0.35 eV, which can be ascribed to the lower electronegativity value of Fe (1.83) compared to Co (1.88), leading to a weaker interaction between Fe and positively charged H atoms[38]. +0.15 eV is also smaller than +0.24 eV, manifesting that the proceeding of the HER process on the Fe-CoS$_2$ surface is easier than on the CoS$_2$, consistent with the experimental results in Fig. 2a, b. This lower value of +0.15 eV than +0.24 eV and thus the HER improvement should be caused by the enhanced charge density of the Co site through the presence of the adjacent Fe atom[39], which can be verified with X-ray absorption near edge structure (XANES) measurements. In the obtained XANES spectra of Co K-edge (Supplementary Fig. 30), the white-line intensity of Fe-CoS$_2$ is clearly lower than that of CoS$_2$, indicating that the Co cations in Fe-CoS$_2$ have been partly reduced[40]. That is, the Co

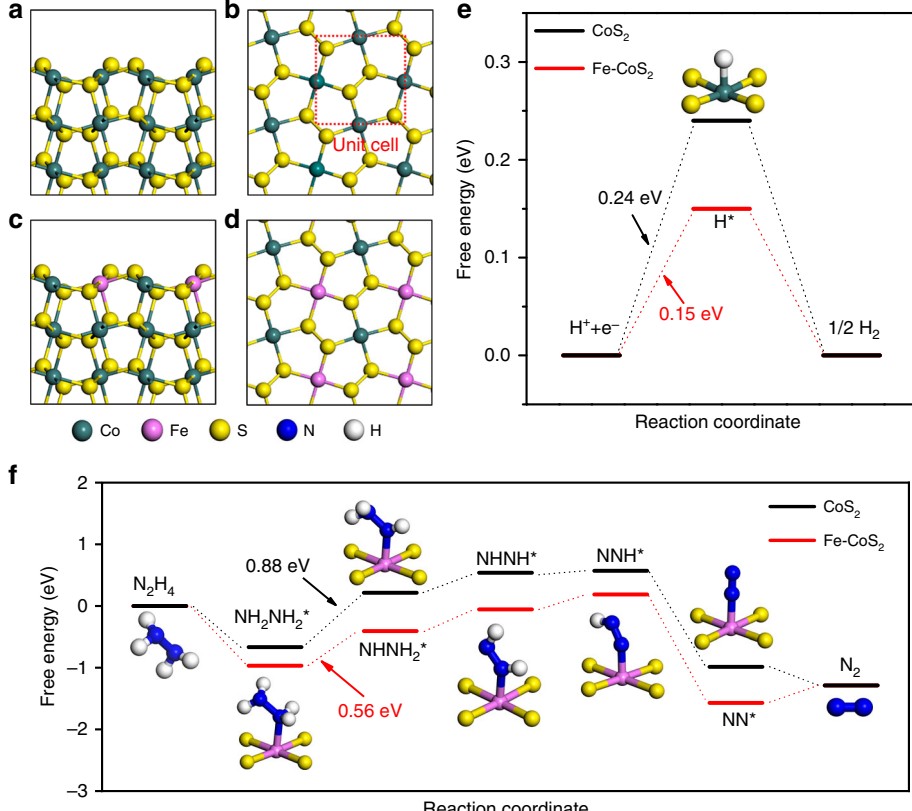

**Fig. 5** Atomic structure models of DFT-relaxed pure $CoS_2$ (001) and Fe-$CoS_2$ (001) surfaces and their calculated profiles of free energy for HER and HzOR. **a**, **b** Side- and top-view images of the $CoS_2$ surface. **c**, **d** Side- and top-view images of the Fe-$CoS_2$ surface. **e** Free energy profiles of HER on the $CoS_2$ and the Fe-$CoS_2$ surfaces at 0 V vs RHE. The inset in **e** is the most stable configuration of the intermediate (H*) adsorbed on the surface of $CoS_2$. The most stable H* configuration on the Fe-$CoS_2$ surface is very close to the one in the inset (see details in Supplementary Fig. 29). **f** Free energy profiles of HzOR on the $CoS_2$ and the Fe-$CoS_2$ surfaces at 0 V vs RHE. The insets in **f** are the $N_2H_4$ and $N_2$ molecular models and the most stable configurations of the adsorbed intermediates on the Fe-$CoS_2$ surface (see more details in Supplementary Fig. 32). Please see Supplementary Fig. 34 for the most stable configurations of the adsorbed intermediates on the $CoS_2$ surface. The numbers with eV in **e**, **f** are the maximum values of positive changes of free energies in the corresponding HER and HzOR processes. The reason to choose (001) is that Fig. 1a indicates the preferential orientation of the nanosheets to be <001>

atoms have got electrons through the presence of the adjacent Fe atoms.

The HzOR processes over the $CoS_2$ (001), the Fe-$CoS_2$ (001) and $FeS_2$ (001) surfaces were also calculated. Figure 5f and Supplementary Figs. 32–34 show that the PDS in each HzOR process is the dehydrogenation of $NH_2NH_2$* to $NHNH_2$*, conforming to the reported[41]. The free-energy change ($\Delta G$) values of PDS on the Co site of the $CoS_2$ surface, on the Fe site of the $FeS_2$ surface, and on the Co and the Fe sites of the Fe-$CoS_2$ surface were calculated to be +0.88 eV (the Co site of $CoS_2$), +0.90 eV (the Fe site of $FeS_2$), +0.74 eV (the Co site of Fe-$CoS_2$) and +0.56 eV (the Fe site of Fe-$CoS_2$), as displayed in Fig. 5f and Supplementary Figs. 32, 33 and 34. These results indicate that not the Co site but the Fe one is the active site for HzOR on Fe-$CoS_2$, and the proceeding of HzOR on Fe-$CoS_2$ is easier than on $CoS_2$ and $FeS_2$, which is in agreement with the experimental results in Fig. 2d, e and Supplementary Fig. 19. The lowest $\Delta G$ value of +0.56 eV and thus the HzOR improvement should also be because Fe atoms have a lower electronegativity compared to Co atoms, which results in a stronger interaction between Fe and negatively charged N atoms[38]. Moreover, the coexistence of Co and Fe atoms in Fe-$CoS_2$ is important for HzOR, which should originate from that the existence of Co atoms further lowers the electronegativity of Fe atoms and thus further enhances the stronger interaction between Fe and negatively charged N atoms[38].

In summary, the Fe-doped $CoS_2$ nanosheets catalyze HER efficiently and robustly over a wide pH range. Remarkable HER

metrics were obtained, including low $\eta$ values, small Tafel slopes, and high TOF values. The nanosheets also exhibit high HzOR activities with a long-term stability and a small potential of 129 mV to deliver 100 mA cm$^{-2}$. Thus, high-performance DHzFCs and OHzS have been achieved using the nanosheets. Further, a DHzFC and an OHzS unit have been integrated to form the first self-powered $H_2$ production system with hydrazine as the sole consumable that serves bifunctionally as the DHzFC fuel and the splitting target. This system works with a hydrogen evolution rate of 9.95 mmol h$^{-1}$, a Faradaic efficiency of 98% and a 20-h stability at room temperature and ambient pressure. These values are all among the best results reported, suggesting that the system is a promising approach without external power sources to generate $H_2$ for industries and energy supply. The above efficient performances are revealed by theoretical calculations to originate from that the Fe doping decreases the free-energy changes of the H adsorption and the $NH_2NH_2$* dehydrogenation on $CoS_2$.

## Methods
**Materials synthesis**. To synthesize thin Fe-$CoS_2$ nanosheets, 0.2 mmol Fe $(NO_3)_2 \cdot 6H_2O$, 5 mmol sodium dodecyl sulfate, 1 mmol $Co(NO_3)_2 \cdot 6H_2O$, and 6 mmol hexamethylenetetramine were dissolved in distilled water (22 mL) at room temperature, giving a uniform solution. This solution was placed into a Teflon-lined autoclave (100 mL) and then heated at 120 °C for 24 h, producing layered hydroxide precursor (for more details, see Supplementary Fig. 1). The hydroxide precursor was mixed with formamide, causing the exfoliation of the host layers, and the resultant suspension was ultrasonicated for 12 h and then was centrifuged, in which powders were collected and found to be thin hydroxide

nanosheets (Supplementary Figs. 2 and 3). The powders were cleaned several times using distilled water and ethanol and afterwards were dried in vacuum at 60 °C for 6 h. After that, the hydroxide precursor and 20 mg of sulfur powder were placed at two different positions in a porcelain boat with the sulfur powder at the upstream region of the furnace and then heated at 350 °C for 3 h under Ar. As a result, thin Fe-CoS$_2$ nanosheets were obtained. Thin CoS$_2$ nanosheets were also made by the same process as the above, except that no Fe(NO$_3$)$_2$·6H$_2$O was added.

For comparison, pure FeS$_2$ nanosheets (Supplementary Fig. 18) were synthesized by directly pyrolyzing $\Upsilon$-FeOOH nanosheets with sulfur powder under Ar at 300 °C for 2 h. The $\Upsilon$-FeOOH nanosheets were synthesized by a hydrolysis process:[31] Specifically, NH$_4$HCO$_3$ (1.50 g) and FeCl$_3$ (0.27 g) were first added to 30 mL propanediol. After 30-min stirring, a homogeneous solution was attained, and it was placed into a Teflon-lined stainless steel autoclave (50 mL). The autoclave was heated at 175 °C for 24 h, leading to the formation of the FeO$_x$–propanediol precursor. Afterward, the precursor was collected, cleaned with deionized water and ethanol by centrifugation several times, and then dried at 60 °C. Further, NaOH (24 mg) and the precursor (20 mg) were added to 60 mL distilled water in a flask with a round bottom, and the obtained mixture was sonicated for 20 min and then heated and kept at 60 °C for 4 h. After this, the precipitates in the mixture were cleaned using deionized water and ethanol for several times and then dried at 60 °C, leading to the achievement of $\Upsilon$-FeOOH nanosheets.

**Characterizations**. XRD patterns were obtained with an X-ray diffractometer, whose model is Rigaku D/max 2500, at a scan rate of 10 degrees min$^{-1}$ in the 10–80° 2$\theta$ range. TEM observations were performed using one conventional TEM (FEI Talos F200X). HAADF-EELS mapping was acquired using an aberration-corrected scanning TEM (STEM) with the model of Titan Cubed Themis G2, whose acceleration voltage was set at 200 kV. AFM measurements were performed with a Veeco Innova equipped with a Si tip. The ICP-MS measurements were conducted with an ICP Perkin-Elmer Optima 3000 DV. Raman measurement was carried out on a Raman microscope (Horiba Jobin Yvon LabRAM HR Evolution), whose excitation wavelength was 532 nm and laser spot size was 1 μm. The N$_2$-sorption isotherms, as well as pore size distribution curves of Fe-CoS$_2$ and CoS$_2$, were obtained by the BET measurement performed on an Autosorb-iQ-MP Micromeritics analyzer. The XANES measurements at the Co K edge were carried out in the fluorescence mode at the XRD station of 4B9A beamline of Beijing Synchrotron Radiation Facility (BSRF).

**Electrochemical measurements**. Firstly, catalysts to be measured were dispersed in a mixture solution, which contained ultrapure water (0.5 mL), alcohol (0.45 mL) and 5 wt.% Nafion (50 μL). After this, the solution was ultrasonicated for 30 min, and it became a homogeneous suspension. This suspension was employed as the catalyst ink. Then, the ink (20 μL) was deposited on an electrode of glassy carbon, causing a Pt loading of 40 μg$_{Pt}$ cm$^{-2}$ for the 20 wt.% Pt/C sample or a loading of 20 μg cm$^{-2}$ for our samples. All the electrochemical measurements for HzOR and HER were carried out at 25 °C in a three-electrode system with different electrolytes and an electrochemical workstation (CHI 760E). A graphite plate was employed as the counter electrode. The reference electrode was Hg/HgO for measurements in 1.0 M KOH, Hg/HgSO$_4$ for measurements in 0.5 M H$_2$SO$_4$, and Hg/Hg$_2$Cl$_2$ for measurements in 1.0 M PBS, respectively. Before HER testing, H$_2$ was injected into the electrolyte solution for 30 min to saturate it with H$_2$. The ohmic drop (namely $iR$ drop) correction was carried out using the current interrupt method with the potentiostat of the workstation[3], which also gave the solution resistances to be 7.2 Ω (1.0 M KOH), 8.9 Ω (0.5 M H$_2$SO$_4$), 14.0 Ω (1.0 M PBS) and 7.8 Ω (1.0 M KOH with 0.1 M hydrazine that is for the HzOR measurement, whose more experimental details will be given later), respectively. These values were consistent with the results from EIS (Supplementary Fig. 35). HER polarization curves were collected with the sweep rate of 1 mV s$^{-1}$ (ref. [15]). TOF values were measured and calculated according to the previously reported equation[6,16]: TOF = $j$ / (2$Fn$), in which $j$ is the HER current density, $n$ is the number of active sites, and $F$ is Faraday constant. The $n$ values were measured and computed by a widely used method[29,30]: Cyclic voltammetry (CV) measurements were performed in the potential range of 0–0.6 V vs RHE when the scan rate was fixed at 50 mV s$^{-1}$ (see obtained CV curves and more details in Supplementary Fig. 16). After this, by integrating the charge of each CV curve over the whole potential range, the half value of the charge was obtained, which is the value of the surface charge density ($Q_s$). Then, the $n$ value was computed by $n = Q_s$ / $F$. Therefore, the $n$ values of Fe-CoS$_2$ were obtained to be 7.54 × 10$^{-8}$ mol cm$^{-2}$ (1.0 M KOH), 5.51 × 10$^{-8}$ mol cm$^{-2}$ (0.5 M H$_2$SO$_4$) and 6.67 × 10$^{-8}$ mol cm$^{-2}$ (1.0 M PBS). EIS spectra were taken with the CHI 660D electrochemical workstation in 1.0 M KOH at an open-circuit potential state in a frequency range (0.1 Hz to 100 kHz), for which the alternating current (AC) voltage amplitude was 5 mV. The ECSA values of electrocatalysts were derived from the $C_{dl}$ values of catalytic surfaces[3,9,16]. In details, we firstly performed CV tests at different scan rates (Supplementary Fig. 11a, b, d and e). Then, plots between the scan rates and the capacitive current densities ($\Delta j = j_{anodic} - j_{cathodic}$) from the CV curves were made (Supplementary Fig. 11c, f), and linear fitting performed on the plots gave the $C_{dl}$ values, from which the ECSA values were derived[3,9,16]. The gas chromatography experiments to measure the amount of generated hydrogen were conducted on a PE5800GC gas

chromatograph containing a thermal conductivity detector. For HzOR, all of the polarization curves were measured in 1.0 M KOH with 0.1 M hydrazine when the scan rate was set at 5 mV s$^{-1}$ (ref. [16]), and the other conditions were the same as the above.

All the electrochemical measurements for OHzS with external power supplies were performed at 25 °C in a two-electrode electrolyzer (namely an OHzS unit; see Supplementary Movie 1 for its optical image) with 1.0 M KOH and 0.1 M hydrazine and the electrochemical workstation. For each pair of OHzS electrodes, the anode and the cathode were prepared by dropping the corresponding catalyst ink on two Ni foams (the catalyst loading was 0.5 mg cm$^{-2}$). Ni foams without catalysts were also tried for OHzS and were found to give a negligible contribution to the observed OHzS activity of Fe-CoS$_2$ supported on Ni foams (see details in Supplementary Fig. 36).

The DHzFC assembly followed the method reported in our previous work[23,24], in which Nafion 115 membranes were used as the solid electrolyte. In three H$_2$O$_2$ DHzFCs, the anodes were the Fe-CoS$_2$ nanosheets, the CoS$_2$ nanosheets, and commercial 40 wt.% Pt/C catalysts, respectively. Their cathodes were all Pt/C (40 wt.%) loaded on carbon paper. For each H$_2$O$_2$ DHzFC testing, an aqueous solution containing 4 M KOH and 20 wt.% N$_2$H$_4$ was added into the anode side with a flow rate of 5 mL min$^{-1}$ by a silicone tube and a peristaltic pump. Meantime, the liquid fed on the cathode side was an acidic solution, which contained 20 wt.% H$_2$O$_2$ and 0.5 M H$_2$SO$_4$, with the same flow rate (5 mL min$^{-1}$) by another peristaltic pump. The cell temperature was controlled by a temperature controller, and its value was monitored by thermocouples. The steady-state $j$–$V$ curves were collected by an automatic electric Load, whose model is PLZ 70UA, Japan. For each O$_2$ or air DHzFC, its assembly and testing were the same as the above, except that the liquid fed on the cathode side was 0.5 M H$_2$SO$_4$ bubbled with O$_2$ or air. A self-powered OHzS system was made using electric cables to connect the electrodes of an O$_2$ DHzFC and an OHzS unit, as shown in Supplementary Movie 2, in which no external power supplies (such as an electrochemical workstation) was used.

**Computational details**. All calculations were performed with dispersion-corrected DFT in the DMol3 code. The Perdew-Burke-Ernzenhof functional was used to treat the generalized gradient approximation. The double numerical basis sets plus the polarization was employed to perform all-electron calculations. The inner electrons were treated by an effective core potential. The Tkatchenko-Scheffler van der Waals correction was used in the dispersion-corrected scheme. The spin-unrestricted method was used for all calculations. The energy convergence tolerance was 2 × 10$^{-5}$ Ha. The maximum displacement and force were 0.005 Å and 0.004 Ha Å$^{-1}$, respectively. The cut-off radius of real-space global orbital was set to be 0.45 nm. The lattice constant of CoS$_2$ was optimized to be 5.580 Å. The vacuum space was set as 1.5 nm for avoiding interactions between periodic images. For geometry optimization, the Brillouin zone was sampled by 5 × 5 × 1 k-points. The binding energy of adsorbents and the free energies in each pathway of electrochemical reaction were calculated on the basis of the computational hydrogen electrode model, which was proposed by Nørskov's group[36]. The $\Delta G$ values on various surfaces were calculated with the equation[36]: $\Delta G = \Delta E + \Delta ZPVE + \int C_p dT - T\Delta S$, where $\Delta E$ is the adsorption energy change, $\Delta ZPVE$ is the difference of the zero-point vibrational energy (ZPVE) between the gas phase and the adsorbed state, and $\Delta S$ is the difference of the entropy between the gas phase and the adsorbed state. $T$ is the system temperature. $C_p$ is the heat capacity. The thermodynamic properties of molecules in the adsorbed state and the gas phase were calculated using vibrational analysis.

## Data availability
The data that support the findings of this study are available within the article (and its Supplementary Information files) and from the corresponding authors upon reasonable request.

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

## Acknowledgements

This work was financially supported by National Key R&D Program of China (2017YFA0700104), National Natural Science Foundation of China (21601136, 21603161 and 51761165012), National Program for Thousand Young Talents of China, Tianjin Municipal Education Commission, and Tianjin Municipal Science and Technology Commission (15JCYBJC52600). The authors would like to thank Dr. Hui Yang, Dr. Wei Xi, Dr. Lili Han, Ms. Jing Shi, and Mr. Ketao Zang for their helps in operating AFM and TEM. The authors also acknowledge National Supercomputing Center in Shenzhen for providing the computational resources and materials studio (version 7.0, DMol3).

## Author contributions

X.J.L. designed the electrocatalysts, performed the synthetic, electrochemical and device experiments, and analyzed the electrochemical data. J.H. performed the calculations and analyzed their results. S.Z.Z. and Y.P.L. performed the XANES measurements and analyzed the results. Z.Z. assisted in the synthetic and electrochemical experiments. X.J.L., J.L., G.Z.H. and Y.D. co-performed the mechanism analysis and co-wrote the paper, to which J.H. and X.M.S. contributed. X.J.L. and J.L. co-proposed the concept of the self-powered hydrogen production system from hydrazine. J.L., G.Z.H. and Y.D. co-supervised this project. All authors discussed the results.
