## [Peer Review File · Nature Communications]

Reviewers' Comments:

Reviewer #1:

Remarks to the Author:

This manuscript reported a Fe-doped CoS₂ nanosheets which are active for HER and HzOR. Fe doping decreases the free-energy changes of H adsorption and adsorbed NH₂NH₂ dehydrogenation on CoS₂. This is a piece of systematic work, however, the Fe-CoS₂ material and so-called self-powered H₂ production system are not novel, and some major inconsistencies exist in experimental data and result discussion. Even the so-called self-powered H₂ production system is just simply assembled by a fuel cell and electrolytic cell using wires, hence, I recommend to reject it or publish it on other journal.

1. In Figure 1f, the d-spacing is inconsistent with the crystal indices of CoS₂ (PDF:89-1492). In Figure 2c, chronoamperometric curve was recorded at potential of -0.2 V for HER, but in Figure 2a, there is no any information about current value at -0.2 V for Fe-CoS₂. The HER polarization curves of the Fe-CoS₂ in Figure 2a and Figure 2c are discrepant. The correction of ohmic potential drop loss from solution resistance was applied to all initial data, please implement the EIS experiment and supply the value of solution resistance.
2. As we know, Pt/C is the best commercial catalyst for HER, but the Fe-CoS₂ nanosheets exhibit better HER activities than Pt/C at large j, this phenomenon should be elucidated clearly. Authors claimed that the Fe-CoS₂ nanosheets exhibit better HER activities than Pt/C at large current and it is important to evaluate intrinsic HER activities using turnover frequency (TOF), please give the TOF of Pt/C. $TOF = j / (2Fn)$, where j is the HER current density, F is Faraday constant and n is the number of active sites. Please supply the computing method and value of n (number of sites) for Fe-CoS₂.
3. The Fe-CoS₂ nanosheets also efficiently catalyzed HzOR in 0.1 M hydrazine, but early reported results are tested in different hydrazine concentrations (Adv. Sci. 4, 1600179 (2017), Adv. Mater. 27, 2361-2366 (2015)), please provide HzOR polarization curves of Fe-CoS₂ in different concentrations (such as 0.33 M, 0.5 M, 1 M). Fe-CoS₂ nanosheets need about -100 mV (Figure 2a) and 129 mV (Figure 2d) to drive 100 mA current of HER and HzOR, but voltage (@100mA cm⁻²) of Fe-CoS₂ nanosheets (+ and -) is 0.6 V in Figure 4a, please explain this discrepancy.
4. The HER improvement should be caused by the enhanced charge density of the Co site through the presence of the adjacent Fe atom (in DFT part), please corroborate this argument by experimental data (such as XANES). Fe is the active site for HzOR on Fe-CoS₂, and the proceeding of HzOR on Fe-CoS₂ is easier than on CoS₂, please supplement the HzOR data of Fe compound without Co.
5. For each pair of OH₂S electrodes, the anode and the cathode were made by dropping the corresponding catalyst ink on two Ni foams (with a catalyst loading of 0.5 mg cm⁻²). Ni foams without catalysts were also tried for OH₂S, and no catalytic currents were observed. However, it is known that NiFe compound is active for anodic oxidation reaction, please correct that and supply the OH₂S data of Ni foam. For hydrogen evolution or hydrazine oxidation, there are many reported samples about CoS₂ (J. Am. Chem. Soc. 2015, 137, 7448-7455, New J. Chem., 2017, 41, 4754-4757), FeCoS_x (J. Am. Chem. Soc. 2015, 137, 4, 1587-1592), so this Fe-CoS₂ material for HER and HzOR is short of novelty.
6. Authors proposed a kind of so-called self-powered H₂ production system with hydrazine. In fact, this system is just simply integrated by DH₂FC and OH₂S unit using wires. Specifically, fuel cell part could be substituted by any other fuel cell (methanol or ethanol) even solar cell, AA battery or other power source. Hence, for novelty of this H₂ production system, authors should provide more original idea rather than confuse concept.

Reviewer #2:

Remarks to the Author:

In this manuscript, Liu et al. report a novel bifunctional electrocatalyst, Fe-doped CoS₂ (Fe-CoS₂) nanosheets, which have the ability to efficiently and robustly catalyze the hydrogen evolution

reaction (HER) and the hydrazine oxidation reaction (HzOR) in alkaline electrolytes. Thus, the authors used Fe-CoS₂ for overall hydrazine splitting (OH₂S), whose performance was found to depend strongly on the Fe doping ratio in the CoS₂. The authors optimized the Fe doping ratio and found that the best ratio was 5.1 at.%, with which Fe-CoS₂ exhibited an outstanding HER activity with a low overpotential of 40 mV for 10 mA cm⁻² and a superior HzOR performance with a low working potential of 129 mV to afford 100 mA cm⁻². These results are both comparable to those of the best reported bifunctional electrocatalysts. Also, when this newly developed catalyst was used as the anode of a direct hydrazine fuel cell (DH₂FC), the resultant DH₂FC exhibited prominent electrochemical performances. More importantly, it is demonstrated that the bifunctional electrocatalyst was successfully used for electrocatalytic OH₂S in a self-powered manner with a DH₂FC under ambient conditions. This newly developed self-powered H₂ production system showed an H₂ evolution rate of 9.95 mmol h⁻¹, surpassing previously reported self-powered H₂ production systems based on water splitting. Moreover, the new system needs only one consumable, hydrazine, in contrast to the two (water and Zn/mechanical/thermal/solar energies) in the previously reported self-powered H₂ production systems with water splitting. This will be highly beneficial to practical applications. Overall, this work exhibits a promising strategy for H₂ production by coupling a DH₂FC and an OH₂S unit.

In addition, it is impressive that Fe-CoS₂ can also catalyze HER under acidic and neutral media. It not only has Pt-like HER activity with small overpotentials but also demonstrates excellent stability as well as nearly 100% Faradaic efficiencies. Furthermore, the authors used theoretical studies to discover that Fe doping decreased the free-energy changes of H adsorption and adsorbed NH₂NH₂ dehydrogenation on CoS₂, which is the scientific reason for the above efficient performances. I think that this work is well done, and the results and explanations are all convincing. This work is the first to present an innovative integration on the combination of a DH₂FC and an OH₂S unit, which is a paradigm for self-powered H₂ production, and thus this manuscript will attract much attention from readers working on energy science and systems, materials science and catalysis. It should be published on Nature Communications, only after the following minor issues are addressed:

1. Figure 2c, f shows the HER and HzOR stabilities of Fe-CoS₂ by chronoamperometric measurements at constant working potentials. For better understanding the stabilities, I suggest that the corresponding TEM images of Fe-CoS₂ after the chronoamperometric measurements should be given.
2. Also regarding to the stabilities, I think the authors should give electrochemical impedance spectroscopy results of Fe-CoS₂ before and after the chronoamperometric stability tests.
3. It is well known that the electrocatalytic processes of HER and HzOR are both typical surface catalytic reactions. Therefore, the specific surface areas and corresponding pore size distributions of Fe-CoS₂ and pure CoS₂ should be provided.
4. Please add the electrochemically active surface areas (ECSAs) of Fe-CoS₂ and CoS₂, as they are also important for the electrocatalysis.

Reviewer #3:

Remarks to the Author:

This manuscript reports Fe-doped CoS₂ nanosheets as efficient bifunctional catalysts for HER and HzOR. Direct hydrazine fuel cells (DH₂FCs) and overall-hydrazine-splitting units (OH₂S) using the developed catalyst have shown remarkable efficiency and stability. The authors further integrated the DH₂FC with the OH₂S unit to form a self-powered H₂ production system, which again showed some of the best H₂ production efficiencies. The work is novel and is appealing to the broad research community. I recommend its publication with the following suggested minor revisions.

- 1) The catalyst loading varied significantly in different studies (20ug/cm² for HER, HzOR, 0.5 mg/cm² for OH₂S, 1.5mg/cm² for DH₂FC). Please provide some justification/ rationale on the selection of different catalyst loadings.

- 2) Based on Fig. 4d, the H₂ production rate appears to be around 0.5 mmol/h, not the claimed 9.95 mmol/h.
- 3) For DH₂FC, O₂ was introduced to the cathode side through "bubbling". How about bubbling air directly?
- 4) While hydrazine can be a potential fuel, its high toxicity, instability etc. can be major hurdles to overcome for its wide-spread applications. It is hard to imagine that automobiles on the street would each carry a tank of hydrazine! Please comment on the viability of hydrazine as a common fuel.

Responses to the Comments

Responses to the comments of Reviewer #1:

Comment:

This manuscript reported a Fe-doped CoS₂ nanosheets which are active for HER and HzOR. Fe doping decreases the free-energy changes of H adsorption and adsorbed NH₂NH₂ dehydrogenation on CoS₂. This is a piece of systematic work, however, the Fe-CoS₂ material and so-called self-powered H₂ production system are not novel, and some major inconsistencies exist in experimental data and result discussion. Even the so-called self-powered H₂ production system is just simply assembled by a fuel cell and electrolytic cell using wires, hence, I recommend to reject it or publish it on other journal.

1. In Figure 1f, the *d*-spacing is inconsistent with the crystal indices of CoS₂ (PDF:89-1492). In Figure 2c, chronoamperometric curve was recorded at potential of –0.2 V for HER, but in Figure 2a, there is no any information about current value at –0.2 V for Fe-CoS₂. The HER polarization curves of the Fe-CoS₂ in Figure 2a and Figure 2c are discrepant. The correction of ohmic potential drop loss from solution resistance was applied to all initial data, please implement the EIS experiment and supply the value of solution resistance.

Response:

Thank you very much for reviewing our manuscript. We greatly appreciate your helpful and constructive comments. Each *d*-spacing value labeled in Figure 1f was obtained by measuring its corresponding lattice distance for six times on the HRTEM image in Figure 1f. For instance, six measurements gave the *d*₂₀₀ value to be 0.275, 0.278, 0.280, 0.279, 0.277 and 0.277 nm, respectively. These six values have the average of 0.278 nm and the standard deviation of 0.002 nm. That is, *d*₂₀₀ = 0.278 ± 0.002 nm. Within the error bars, this value equals the theoretical one (0.2769 nm) of {200} from PDF: 89-1492. Similarly, the *d*₀₁₂ value was measured to be 0.247 ± 0.001 nm, which also equals the theoretical value (0.2477 nm) of {012} from PDF: 89-1492 within the error bars. 0.278 and 0.247 can be rounded to 0.28 and 0.25, respectively. This is the reason why we labeled 0.28 nm and 0.25 nm for *d*₂₀₀ and *d*₀₁₂, respectively, in Figure 1f. Your comment lets us realize that we should label these values more accurately, and thus we have changed 0.28 nm and 0.25 nm to 0.278 ± 0.002 nm and 0.247 ± 0.001 nm, respectively, in Figure 1f and revised the corresponding texts (please see them in Page 6 of the main text). For your convenience, all changes made for the responses have been highlighted by yellow in the main text and the Supplementary Information (SI) files.

In the original version of Figure 2a, the scale of the Y axis was not long enough to show the current density value at –0.2 V. We are really sorry for this and have extended the Y-axis scale in Figure 2a, from which you can now find the current density value at –0.2 V to be 296 mA cm⁻². This value is very close to the onset value (298 mA cm⁻²) of the chronoamperometric curve in Figure 2c. In addition to the revision of Figure 2a, the description of the above content has been added in Page 8 of the main text file.

The initial HER polarization curve of the Fe-CoS₂ measured in 1.0 M KOH should be displayed for two times, of which one is in Figure 2a and the other is in the inset of Figure 2c. That is, the initial HER polarization curves of the Fe-CoS₂ in Figure 2a and the inset of Figure 2c should be identical. According to your comment, we have checked carefully the data through the manuscript and found that we placed the initial HER polarization curve of the Fe-CoS₂ and its curve after 10,000 cycles measured in 1.0 M PBS in the inset of Figure 2c by mistake (the correct position of the two PBS-measured curves is the inset of Supplementary Figure 14f). The reason why this mistake happened is that a plenty of data were presented in the manuscript. We are really sorry for this mistake and have changed the PBS-measured curves to the KOH-measured ones. Now you can find that the initial HER polarization curves in Figure 2a and the inset of Figure 2c are identical. In addition, we have checked and confirmed that such mistakes do not exist in Supplementary Figure 14f and all other figures.

The correction of ohmic potential drop loss from solution resistance was performed using the conventional current-interrupt method with the potentiostat of an electrochemical workstation, which gave the solution resistance values to be 7.2 Ω (1.0 M KOH), 8.9 Ω (0.5 M H₂SO₄), 14.0 Ω (1.0 M PBS) and 7.8 Ω (1.0 M KOH with 0.1 M hydrazine), respectively. According to your comment, we have taken the electrochemical impedance spectra (EIS) in the different electrolytes, and the results are added as Supplementary Figure 35, which give the solution resistance values to be 7.4 Ω (1.0 M KOH), 9.0 Ω (0.5 M H₂SO₄), 13.9 Ω (1.0 M PBS) and 7.9 Ω (1.0 M KOH with 0.1 M hydrazine), respectively. Each of the values is close to the corresponding one obtained by the workstation. In addition to Supplementary Figure 35, the experimental details and description of the above results have been added in Page 30 of the main text file and Page 39 of the SI file.

Comment:

2. As we know, Pt/C is the best commercial catalyst for HER, but the Fe-CoS₂ nanosheets exhibit better HER activities than Pt/C at large j , this phenomenon should be elucidated clearly. Authors claimed that the Fe-CoS₂ nanosheets exhibit better HER activities than Pt/C at large current and it is important to evaluate intrinsic HER activities using turnover frequency (TOF), please give the TOF of Pt/C. $TOF = j / (2Fn)$, where j is the HER current density, F is Faraday constant and n is the number of active sites. Please supply the computing method and value of n (number of sites) for Fe-CoS₂.

Response:

Thank you for your constructive comment and suggestions. Our work has indeed shown the following findings: in 1.0 M KOH (Figure 2a), the Fe-CoS₂ nanosheets exhibited better (or worse) HER activities than Pt/C at large (or small) j ; the result obtained in 1.0 M PBS (Supplementary Figure 14b) is similar to that in 1.0 M KOH; in 0.5 M H₂SO₄ (Supplementary Figure 14a), the HER activities of Fe-CoS₂ were always worse than those of Pt/C at small and large j . In order to elucidate these phenomena, we have used EIS to measure the charge transfer resistance (R_{ct}) values of Fe-CoS₂ and Pt/C in the three electrolyte solutions. The results are added as Supplementary Figure 10, which give the R_{ct} values of Fe-CoS₂ to be 24.2 Ω (1.0 M KOH), 35.5 Ω (1.0 M PBS) and 39.2 Ω (0.5 M H₂SO₄). The R_{ct} values of Pt/C are given to be 47.3 Ω (1.0 M KOH), 56.4 Ω (1.0 M PBS) and 25.8 Ω (0.5 M H₂SO₄). Together with Figure 2a and Supplementary Figure 14a,b, the above results indicate the following mechanism: in 1.0 M KOH or 1.0 M PBS, Pt/C

exhibited a typical 0 mV overpotential at the onset position, and its corresponding HER current density started to increase from 0 earlier and faster than that of Fe-CoS₂; but, when the applied potential was increased further, the increase rate of the HER current density of Pt/C was lower than that of Fe-CoS₂, due to the larger charge transfer resistance of Pt/C; thus, after the applied potential was increased to be larger than a critical value (that is, the HER current density was larger than a critical value), the HER current density of Fe-CoS₂ started to be larger than that of Pt/C, as displayed in Figure 2a or Supplementary Figure 14b, showing better HER activities than Pt/C. Similar cases have been reported on other electrocatalysts in the literature, such as MoS₂ (Lu et al. *Adv. Mater.* 2014, 26, 2683) and Ni-Mo alloy (Zhang et al. *Small* 2017, 13, 1701648). In contrast, in 0.5 M H₂SO₄, the charge transfer resistance of Fe-CoS₂ is larger than that of Pt/C, and thus its HER activities were always worse than those of Pt/C. The description of the above content has been added in Page 9 of the main text file and Pages 11 and 17 of the SI file. The papers (Lu et al. *Adv. Mater.* 2014, 26, 2683 and Zhang et al. *Small* 2017, 13, 1701648) have been cited as refs. 27 and 28 in the revised manuscript.

Our initially submitted manuscript used the equation $TOF = j / (2Fn)$ to give the TOF values of Fe-CoS₂ at the overpotential (η) of 100 mV to be 6.74, 4.76 and 2.21 s⁻¹ in 1.0 M KOH, 0.5 M H₂SO₄ and 1.0 M PBS, respectively, for the HER. The details to measure and compute the n values will be given and discussed in the next paragraph. We have also used the equation to obtain the TOF values of Pt/C at $\eta = 100$ mV to be 1.53, 5.46 and 2.05 s⁻¹ in 1.0 M KOH, 0.5 M H₂SO₄ and 1.0 M PBS, respectively. Comparing the two sets of TOF values indicates that the intrinsic HER activities of Fe-CoS₂ are better than those of Pt/C in 1.0 M KOH and 1.0 M PBS but worse than that of Pt/C in 0.5 M H₂SO₄, which is consistent with the results and discussion shown in the previous paragraph. The description of the above content has been added in Pages 10 and 11 of the main text file.

In our manuscript, the n values were measured and computed by a method widely used in the literature, such as Merki et al. *Chem. Sci.* 2011, 2, 1262 and Xu et al. *Energy Environ. Sci.* 2018, 11, 1819. Its details are as follows: Cyclic voltammetry (CV) measurements were conducted in the potential range of 0 – 0.6 V vs RHE at a fixed scan rate of 50 mV s⁻¹ (the obtained CV curves are added as Supplementary Figure 16). After this, by integrating the charge of each CV curve over the whole potential range, the half value of the charge was obtained, which is the value of the surface charge density (Q_s). Then, the n value was computed by $n = Q_s/F$, where F is the Faraday constant. Thus, the n values of Fe-CoS₂ were obtained to be 7.54×10^{-8} mol cm⁻² (1.0 M KOH), 5.51×10^{-8} mol cm⁻² (0.5 M H₂SO₄) and 6.67×10^{-8} mol cm⁻² (1.0 M PBS). The description of the above content has been added in Pages 10, 11 and 30 of the main text file and Pages 19 and 20 of the SI file. The papers (Merki et al. *Chem. Sci.* 2011, 2, 1262 and Xu et al. *Energy Environ. Sci.* 2018, 11, 1819) have been cited as refs. 29 and 30 in the revised manuscript.

Comment:

3. The Fe-CoS₂ nanosheets also efficiently catalyzed HzOR in 0.1 M hydrazine, but early reported results are tested in different hydrazine concentrations (Adv. Sci. 4, 1600179 (2017), Adv. Mater. 27, 2361–2366 (2015)), please provide HzOR polarization curves of Fe-CoS₂ in different concentrations (such as 0.33 M, 0.5 M, 1 M). Fe-CoS₂ nanosheets need about -100 mV (Figure 2a) and 129 mV (Figure 2d) to drive 100 mA current of HER and HzOR, but voltage (@100mA cm⁻²) of Fe-CoS₂ nanosheets (+ and -) is 0.6 V in Figure 4a, please explain this discrepancy.

Response:

Thank you for these helpful suggestions and comment, according to which we have taken the HzOR polarization curves of Fe-CoS₂ in the hydrazine concentrations of 0.33, 0.5 and 1 M. The curves are added as Supplementary Figure 20. Together with the curve in 0.1 M hydrazine (Figure 2d), they show that higher hydrazine concentrations correspond to larger HzOR current densities in the main range of applied potentials. This observation further confirms that the Fe-CoS₂ nanosheets are efficient for catalyzing the HzOR. The description and experimental details of the above content have been added in Page 11 of the main text file and Page 24 of the SI file. The papers of Feng et al. *Adv. Sci.* 2017, 4, 1600179 and Lu et al. *Adv. Mater.* 2015, 27, 2361 had been cited as refs. 24 and 23, respectively, in our initially submitted manuscript, and their numbering is unchanged in the revised manuscript.

In the individual HER (Figure 2a) and HzOR (Figure 2d) measurements of Fe-CoS₂, we used the typical three-electrode system to carry out the measurements, in which an Hg/HgO electrode was utilized as the reference electrode. All of the potentials in the HER and HzOR polarization curves in Figure 2a,d are values relative to the reference electrode, which can be converted to values versus reversible hydrogen electrode (RHE). Overall hydrazine splitting (OH₂S) measurements are different, in which a two-electrode system was employed (Figure 4a). For the OH₂S of Fe-CoS₂, the two working electrodes of the two-electrode system were both Fe-CoS₂. Thus, the voltages reported by the system reflect the real driving forces between the two Fe-CoS₂ electrodes, which are not the relative potentials. That is, "potential" and "voltage" are two different concepts here, and thus their values do not equal each other. Such cases have been reported on other electrocatalysts in the literature, such as Ni-Co complexes hybridized with 1T-phase MoS₂ (ref. 9 cited in our manuscript) and Zn-doped Co₃O₄ sheets (ref. 15). The description of the above content has been added in Page 17 of the main text file.

Comment:

4. The HER improvement should be caused by the enhanced charge density of the Co site through the presence of the adjacent Fe atom (in DFT part), please corroborate this argument by experimental data (such as XANES). Fe is the active site for HzOR on Fe-CoS₂, and the proceeding of HzOR on Fe-CoS₂ is easier than on CoS₂, please supplement the HzOR data of Fe compound without Co.

Response:

Thank you for your inspiring suggestions. Our DFT part has indeed given the following discussion and results: the HER improvement should be caused by the enhanced charge density of the Co site through the presence of the adjacent Fe atom; Fe is the active site for HzOR on Fe-CoS₂, and the proceeding of HzOR on Fe-CoS₂ is easier than on CoS₂. To carry out your suggestion, we have invited Dr. Shunzheng Zhao in University of Science and Technology Beijing and Dr. Yunpeng Liu in Institute of High Energy Physics, Chinese Academy of Sciences, to join our research group. They have performed X-ray absorption near edge structure (XANES) measurements on Fe-CoS₂ and pure CoS₂ in Beijing Synchrotron Radiation Facility. The XANES results are added as Supplementary Figure 30, which shows that the intensity of the white line (the white line means the first peak after absorption edge) of Fe-CoS₂ is obviously lower than that of CoS₂. This intensity difference

indicates that the Co cations in Fe-CoS₂ have been partly reduced, according to Zhao et al. *Mater. Chem. Phys.* 2018, 205, 35. That is, the Co atoms have got electrons through the presence of the adjacent Fe atoms. Therefore, the XANES result corroborates the DFT result about the HER. The description and experimental details of the above content have been added in Pages 19 and 29 of the main text file. The paper (Zhao et al. *Mater. Chem. Phys.* 2018, 205, 35) has been cited as ref. 40 in the revised manuscript. The names of Drs. Zhao and Liu have been added into the author list, and their contributions have been stated in the section of Author Contributions in Page 27 of the main text file.

According to your suggestion about Fe compound without Co, we have synthesized pure FeS₂ nanosheets by a hydrolysis reaction and a sulfurization process. The hydrolysis reaction is adopted from Yang et al. *Chem. Asian J.* 2014, 9, 1563. The characterization results and synthesis details of the pure FeS₂ nanosheets have been added as Supplementary Figure 18 and into Pages 28 and 29 of the main text file, respectively. The FeS₂ nanosheet thickness was measured by AFM to be 1.31 ± 0.04 nm (Supplementary Figure 18c,d), very close to that of the Fe-CoS₂ nanosheets (1.22 ± 0.03 nm). We have also taken the HzOR polarization curve of the FeS₂ nanosheets under the same conditions as those of Fe-CoS₂ (Figure 2d). The curve is added as Supplementary Figure 19, which shows that the FeS₂ nanosheets exhibited a potential of 428 mV for delivering the current density of 100 mA cm⁻² (namely $E_{100} = 428$ mV). This E_{100} value is obviously larger than those of CoS₂ (205 mV) and Fe-CoS₂ (129 mV), as shown in Figure 2d. This comparison indicates the inferior HzOR activity of FeS₂. To explore the mechanism of the inferior activity, we have performed DFT calculations on FeS₂. The DFT results are added as Supplementary Figure 33, which shows the following findings: the Fe site is the active site for the HzOR on the FeS₂ surface, the potential-determining step (PDS) in the HzOR process on FeS₂ is the dehydrogenation of NH₂NH₂* to NHHH₂*, and the two results are the same as those on Fe-CoS₂ (Figure 5f); but, the free-energy change (ΔG) value of PDS on the Fe site of FeS₂ was calculated to be +0.90 eV, larger than those on the Fe site of Fe-CoS₂ (+0.56 eV) and on the Co site of CoS₂ (+0.88 eV). These results indicate that the proceeding of HzOR on Fe-CoS₂ or CoS₂ is easier than on FeS₂, which is consistent with the experimental results mentioned above, and the coexistence of Co and Fe atoms in Fe-CoS₂ is important for HzOR. This importance should originate from that the presence of Co atoms lowers the electronegativity of Fe atoms, leading to a further stronger interaction between Fe and negatively charged N atoms, according to ref. 38 cited in our manuscript. The description of the above content has been added in Pages 11, 21 and 22 of the main text file. The paper (Yang et al. *Chem. Asian J.* 2014, 9, 1563) has been cited as ref. 31 in the revised manuscript.

Comment:

5. For each pair of OHZS electrodes, the anode and the cathode were made by dropping the corresponding catalyst ink on two Ni foams (with a catalyst loading of 0.5 mg cm⁻²). Ni foams without catalysts were also tried for OHZS, and no catalytic currents were observed. However, it is known that NiFe compound is active for anodic oxidation reaction, please correct that and supply the OHZS data of Ni foam. For hydrogen evolution or hydrazine oxidation, there are many reported samples about CoS₂ (*J. Am. Chem. Soc.* 2015, 137, 7448-7455, *New J. Chem.*, 2017, 41, 4754-4757), FeCoS_x (*J. Am. Chem. Soc.* 2015, 137, 4, 1587-1592), so this Fe-CoS₂ material for HER and HzOR is short of novelty.

Response:

Thank you for your useful comments. We have taken the OHzS polarization curve of the Ni foams under the same conditions as those of the catalysts (Figure 4a). The curve is added as Supplementary Figure 36, which shows that the Ni foams required a cell voltage of 1.20 V to achieve the OHzS current density of 10 mA cm^{-2} . This cell voltage is much higher than that of Fe-CoS₂ (0.002 V) for 10 mA cm^{-2} , which is shown in Figure 4a. Moreover, when the cell voltage was 0.002 V, the current density of the Ni foams was -0.01 mA cm^{-2} , far lower than that of Fe-CoS₂ (10 mA cm^{-2}). That is, if we use the 10 mA cm^{-2} of Fe-CoS₂ as a reference, the -0.01 mA cm^{-2} of the Ni foams is very close to zero. Besides, the negative sign of the -0.01 mA cm^{-2} indicates that the OHzS did not occur at that moment. Therefore, Ni foams gave a negligible contribution to the observed OHzS activity of Fe-CoS₂ supported on Ni foams. The description of the above content has been added in Page 31 of the main text file and Page 40 of the SI file.

Indeed, Kornienko et al. *J. Am. Chem. Soc.* 2015, 137, 7448 reported CoS_x films, which exhibited $\eta = 83 \text{ mV}$ at 2 mA cm^{-2} in 0.5 M PBS for the HER at room temperature; Ma et al. *New J. Chem.* 2017, 41, 4754 reported that CoS₂ nanowire arrays showed a potential of $\sim 210 \text{ mV}$ to achieve 200 mA cm^{-2} in 1.0 M KOH with 0.1 M hydrazine for the HzOR at room temperature; Wang et al. *J. Am. Chem. Soc.* 2015, 137, 1587 reported that Co-doped FeS₂ nanosheets hybridized with carbon nanotubes exhibited $\eta = \sim 120 \text{ mV}$ at 20 mA cm^{-2} in 0.5 M H₂SO₄ for the HER at room temperature. These are all significant researches for CoS₂- and FeS₂-based electrocatalysts. Following them, our Fe-doped CoS₂ nanosheets with the CoS₂ crystal structure unchanged have shown improved performances, such as $\eta = 4 \text{ mV}$ at 2 mA cm^{-2} (1.0 M PBS) and $\eta = 55 \text{ mV}$ at 20 mA cm^{-2} (0.5 M H₂SO₄) for the HER and a potential of 166 mV at 200 mA cm^{-2} (1.0 M KOH with 0.1 M hydrazine) for the HzOR at room temperature. Moreover, we have read through not only the above three papers but also all other papers about CoS₂- and FeS₂-based materials, in all of which no Fe-doped CoS₂ nanosheets with the CoS₂ crystal structure unchanged have been reported. Further, our Fe-CoS₂ nanosheets have been successfully employed as a bifunctional electrocatalyst with high performances for the OHzS. We think that the above features together indicate the novelty of the Fe-CoS₂ nanosheets. The above discussion has been added in Page 12 of the main text file. The papers (Kornienko et al. *J. Am. Chem. Soc.* 2015, 137, 7448, Ma et al. *New J. Chem.* 2017, 41, 4754 and Wang et al. *J. Am. Chem. Soc.* 2015, 137, 1587) have been cited as refs. 32–34 in the revised manuscript.

Comment:

6. Authors proposed a kind of so-called self-powered H₂ production system with hydrazine. In fact, this system is just simply integrated by DHzFC and OHzS unit using wires. Specifically, fuel cell part could be substituted by any other fuel cell (methanol or ethanol) even solar cell, AA battery or other power source. Hence, for novelty of this H₂ production system, authors should provide more original idea rather than confuse concept.

Response:

Thank you for your helpful comments. Indeed, the direct hydrazine fuel cell (DHzFC) in our self-powered H₂ production system with hydrazine can be substituted by a methanol/ethanol fuel cell, a solar cell, an AA battery or other power source. But, if the substitution is done, not only hydrazine but also a second consumable (such as methanol, ethanol, sunlight, or AA battery) will

be needed by the system, which will make the prospective application of the system less convenient. In contrast, the system with a DHzFC needs only one consumable, hydrazine, which serves bifunctionally as the DHzFC fuel and the splitting target, and this concept simplifies the consumable supply and thus the prospective application of self-powered H₂ production systems. Moreover, up to now, the concept of the self-powered H₂ production system with bifunctional hydrazine as a sole consumable has not been reported in the literature, and only five self-powered H₂ production systems using water have been reported by refs. 10–12, 18 and 19 cited in our manuscript. The five groundbreaking systems creatively integrate electrocatalytic overall-water-splitting (OWS) units and Zn–air batteries/nanogenerators/thermoelectric cells/solar cells, and they use two consumables, water and Zn or mechanical/thermal/solar energies, to produce H₂ and O₂ with high H₂ evolution rates of 0.08–8 mmol h⁻¹. Inspired by them, our system transforms the products to H₂ and N₂, reduces two consumables to one, and utilizes the low-overpotential HzOR to achieve the improved H₂ evolution rate of 9.95 mmol h⁻¹. We think that the above features together indicate the novelty of the self-powered H₂ production system with bifunctional hydrazine as a sole consumable. The above discussion has been added in Page 17 of the main text file.

Responses to the comments of Reviewer #2:

Comment:

In this manuscript, Liu et al. report a novel bifunctional electrocatalyst, Fe-doped CoS₂ (Fe-CoS₂) nanosheets, which have the ability to efficiently and robustly catalyze the hydrogen evolution reaction (HER) and the hydrazine oxidation reaction (HzOR) in alkaline electrolytes. Thus, the authors used Fe-CoS₂ for overall hydrazine splitting (OH₂S), whose performance was found to depend strongly on the Fe doping ratio in the CoS₂. The authors optimized the Fe doping ratio and found that the best ratio was 5.1 at.%, with which Fe-CoS₂ exhibited an outstanding HER activity with a low overpotential of 40 mV for 10 mA cm⁻² and a superior HzOR performance with a low working potential of 129 mV to afford 100 mA cm⁻². These results are both comparable to those of the best reported bifunctional electrocatalysts. Also, when this newly developed catalyst was used as the anode of a direct hydrazine fuel cell (DH₂FC), the resultant DH₂FC exhibited prominent electrochemical performances.

More importantly, it is demonstrated that the bifunctional electrocatalyst was successfully used for electrocatalytic OH₂S in a self-powered manner with a DH₂FC under ambient conditions. This newly developed self-powered H₂ production system showed an H₂ evolution rate of 9.95 mmol h⁻¹, surpassing previously reported self-powered H₂ production systems based on water splitting. Moreover, the new system needs only one consumable, hydrazine, in contrast to the two (water and Zn/mechanical/thermal/solar energies) in the previously reported self-powered H₂ production systems with water splitting. This will be highly beneficial to practical applications. Overall, this work exhibits a promising strategy for H₂ production by coupling a DH₂FC and an OH₂S unit.

In addition, it is impressive that Fe-CoS₂ can also catalyze HER under acidic and neutral media. It not only has Pt-like HER activity with small overpotentials but also demonstrates excellent stability as well as nearly 100% Faradaic efficiencies. Furthermore, the authors used theoretical studies to discover that Fe doping decreased the free-energy changes of H adsorption and adsorbed NH₂NH₂ dehydrogenation on CoS₂, which is the scientific reason for the above efficient performances. I think that this work is well done, and the results and explanations are all convincing. This work is the first to present an innovative integration on the combination of a DH₂FC and an OH₂S unit, which is a paradigm for self-powered H₂ production, and thus this manuscript will attract much attention from readers working on energy science and systems, materials science and catalysis. It should be published on Nature Communications, only after the following minor issues are addressed:

1. Figure 2c, f shows the HER and HzOR stabilities of Fe-CoS₂ by chronoamperometric measurements at constant working potentials. For better understanding the stabilities, I suggest that the corresponding TEM images of Fe-CoS₂ after the chronoamperometric measurements should be given.

Response:

Thank you very much for reviewing our manuscript. We greatly appreciate your helpful and constructive comments. The TEM characterizations of Fe-CoS₂ after the HER and HzOR chronoamperometric measurements have been performed, and the obtained TEM images are added as Supplementary Figure 13a,c. Besides, the X-ray diffraction (XRD) patterns of Fe-CoS₂

after the HER and HzOR chronoamperometric measurements have also been taken, and they are added as Supplementary Fig. 13b,d. Figure 1a,b has given the XRD pattern and TEM image of Fe-CoS₂ before the chronoamperometric measurements. Comparing Figure 1a,b and Supplementary Figure 13 shows that the Fe-CoS₂ nanosheets preserve the same structure and morphology before and after the chronoamperometric experiments, indicating their structural stability. In addition to Supplementary Figure 13, the description of the above content has been added in Pages 10 and 12 of the main text file and Page 15 of the Supplementary Information (SI) file. For your convenience, all changes made for the responses have been highlighted by yellow in the main text and the SI files.

Comment:

2. Also regarding to the stabilities, I think the authors should give electrochemical impedance spectroscopy results of Fe-CoS₂ before and after the chronoamperometric stability tests.

Response:

Thank you for your helpful suggestion. The electrochemical impedance spectroscopy (EIS) measurements of Fe-CoS₂ before and after the chronoamperometric stability tests have been performed, and the obtained EIS spectra are added as Supplementary Figure 12, which shows that the two spectra before and after the HER chronoamperometric stability test are almost identical to each other. So are the two before and after the HzOR chronoamperometric stability test. These results indicate that Fe-CoS₂ has stable electrocatalytic kinetics through the stability tests. The description of the above content has been added in Pages 10 and 12 of the main text file and Page 14 of the SI file.

Comment:

3. It is well known that the electrocatalytic processes of HER and HzOR are both typical surface catalytic reactions. Therefore, the specific surface areas and corresponding pore size distributions of Fe-CoS₂ and pure CoS₂ should be provided.

Response:

Thank you for pointing this out. As you suggested, the N₂-sorption isotherms and pore size distribution curves of the Fe-CoS₂ and the pure CoS₂ nanosheets have been added as Supplementary Figure 7. The isotherms give the specific surface area values of Fe-CoS₂ and pure CoS₂ to be 27.2 and 25.3 m² g⁻¹, respectively, which are close to each other. The pore size distribution curves indicate the presence of micropores (<2 nm) and mesopores (2–50 nm) in both of Fe-CoS₂ and pure CoS₂. These findings confirm that the Fe doping did not change the morphology and structure of the nanosheets, which is consistent with the XRD (Figure 1a and Supplementary Figure 4), Raman (Supplementary Figure 5), and HRTEM (Figure 1f) results. The description of the above content has been added in Pages 6 and 29 of the main text file and Page 8 of the SI file.

Comment:

4. Please add the electrochemically active surface areas (ECSAs) of Fe-CoS₂ and CoS₂, as they are also important for the electrocatalysis.

Response:

Thank you for this helpful comment. The ECSAs of CoS₂ and Fe-CoS₂ have been measured by the conventional method of electrochemical double-layer capacitance (C_{dl}), which has been widely used in the literature, such as refs. 3, 9 and 16 cited in our manuscript. In details, firstly we performed cyclic voltammogram (CV) tests at different scan rates, and the obtained CV curves are added as Supplementary Figure 11a, b, d and e. Then, plots between the scan rates and the capacitive current densities from the CV curves were made (Supplementary Figure 11c, f), and linear fittings performed on the plots gave the C_{dl} values, from which the ECSA values of CoS₂ and Fe-CoS₂ were derived to be 683 and 900 cm², respectively, for the HER, and 817 and 1,200 cm², respectively, for the HzOR. The two pairs of ECSA values indicate the presence of more active surface areas on Fe-CoS₂ than on CoS₂, in a good agreement with the better HER and HzOR activities of Fe-CoS₂ than CoS₂ (Figure 2a, d). The description of the above content has been added in Pages 9, 10, 12, 30 and 31 of the main text file and Pages 12 and 13 of the SI file.

Responses to the comments of Reviewer #3:

Comment:

This manuscript reports Fe-doped CoS₂ nanosheets as efficient bifunctional catalysts for HER and HzOR. Direct hydrazine fuel cells (DHzFCs) and overall-hydrazine-splitting units (OHzS) using the developed catalyst have shown remarkable efficiency and stability. The authors further integrated the DHzFC with the OHzS unit to form a self-powered H₂ production system, which again showed some of the best H₂ production efficiencies. The work is novel and is appealing to the broad research community. I recommend its publication with the following suggested minor revisions.

1) The catalyst loading varied significantly in different studies (20 ug/cm² for HER, HzOR, 0.5 mg/cm² for OHzS, 1.5 mg/cm² for DHzFC). Please provide some justification/ rationale on the selection of different catalyst loadings.

Response:

Thank you very much for reviewing our manuscript. We greatly appreciate your helpful and constructive comments. The catalyst loadings you mentioned are all optimized results. For instance, we gradually increased the catalyst loading of Fe-CoS₂ from 5 to 80 μg cm⁻² and measured an HER polarization curve for each loading. The HER results are added as Supplementary Figure 9a,b, showing that when the loading increased, the overpotential value for 10 mA cm⁻² (denoted as η_{10}) initially decreased, reached the lowest point at 20 μg cm⁻² and then increased. This change indicates that the HER performance of Fe-CoS₂ was the highest at 20 μg cm⁻². Thus, we used this loading for the HER experiments in the main text. In addition, the increase of η_{10} after 20 μg cm⁻² was possibly due to the slow electron-transfer kinetics caused by the over-loading of catalysts (Zhao et al. *Small* 2017, 13, 1701519). Likewise, the Fe-CoS₂ loadings for the HzOR, DHzFC and OHzS were also optimized, and the results are added as Supplementary Figures 9c,d, 22 and 24, respectively. They show that the best loadings for the HzOR, DHzFC and OHzS are 20 μg cm⁻², 1.5 mg cm⁻² and 0.5 mg cm⁻², respectively. In addition to Supplementary Figures 9, 22 and 24, the description of the above content has been added in Pages 7, 11, 14 and 15 of the main text file and Pages 10, 26 and 28 of the Supplementary Information (SI) file. The paper (Zhao et al. *Small* 2017, 13, 1701519) has been cited as ref. 26 in the revised manuscript. For your convenience, all changes made for the responses have been highlighted by yellow in the main text and the SI files.

Comment:

2) Base on Fig. 4d, the H₂ production rate appears to be around 0.5 mmol/h, not the claimed 9.95 mmol/h.

Response:

Thank you for pointing this out. We also optimized the hydrazine concentration for the H₂ production rate during the preparation of the initially submitted version of our manuscript, and the results are now added as Supplementary Figure 27a, which shows that when the hydrazine concentration increased from 0.1 to 5.6 M, the H₂ production rate exhibited an initial value of 0.47 mmol h⁻¹, then increased and finally became saturated at 5.3–5.6 M. The saturated value is 9.95 mmol h⁻¹. The original version of Figure 4d gave only the generated amounts of H₂ and N₂ corresponding to the initial value of 0.47 mmol h⁻¹. We are really sorry for this, and we have

moved the original version of Figure 4d to Supplementary Figure 27b and displayed the generated amounts of H₂ and N₂ corresponding to 9.95 mmol h⁻¹ and 5.3 M in the new version of Figure 4d. In addition, the description of the above content has been added in Pages 16 and 17 of the main text file and Page 31 of the SI file.

Comment:

3) For DHzFC, O₂ was introduced to the cathode side through "bubbling". How about bubbling air directly?

Response:

Thank you for this constructive comment, according to which we have re-performed the DHzFC measurement by bubbling air directly. Except the air bubbling, the other conditions of the measurement are all the same as those of the O₂ DHzFCs (Figure 3d). The new measurement results are added as Supplementary Figure 23, which shows that the air DHzFCs worked. Moreover, the results give the maximum power density (P_{\max}) values of the air DHzFCs with Fe-CoS₂, CoS₂ and Pt/C to be 35, 13 and 21 mW cm⁻², respectively. These P_{\max} values are much lower than those of the O₂ DHzFCs (125, 36 and 66 mW cm⁻²). This is because air is not pure O₂. The description and experimental details of the above content have been added in Pages 14 and 31 of the main text file.

Comment:

4) While hydrazine can be a potential fuel, its high toxicity, instability etc. can be major hurdles to overcome for its wide-spread applications. It is hard to image that automobiles on the street would each carry a tank of hydrazine! Please comment on the viability of hydrazine as a common fuel.

Response:

Thank you for pointing this out. Indeed, the high toxicity and instability of hydrazine can be major hurdles to overcome for its wide-spread applications. Fortunately, Asazawa et al. *Angew. Chem. Int. Ed.* 2007, 46, 8024 has proposed a detoxification technique to deal with the problems. Its details are as follows: a hydrazine-fixing polymer can be synthesized by mixing sodium 4-vinylbenzene sulfonate and methyl vinyl ketone and cross-linking them with methylene(bis)acrylamide; the polymer contains carbonyl (>C=O) groups; the groups can react with hydrazine to form hydrazone (>C=N-NH₂) groups, which are harmless and stable on the polymer; when hydrazine is needed, hydrazine molecules can be released from the hydrazone groups by placing the polymer in water or a KOH aqueous solution. The polymer and KOH aqueous solution are both reusable, and so they are not consumables. Moreover, the polymer is harmless and stable, and KOH aqueous solutions containing hydrazine are already used in our self-powered system. Therefore, the detoxification technique should be an effective method to overcome the high toxicity and instability of hydrazine. The description of the above content has been added in Pages 17 and 18 of the main text file. The paper (Tanaka et al. *Angew. Chem. Int. Ed.* 2007, 46, 8024) has been cited as ref. 35 in the revised manuscript.

Reviewer #1:

Remarks to the Author:

The paper is greatly improved, so I suggest it to be accepted for publication in its present form.

Reviewer #2:

Remarks to the Author:

I read it carefully and found that the authors have made a commendable effort to address all the concerns of mine and the other reviewers'. Firstly, the authors have fully complemented the stability confirmation, stability origins and surface information of the catalyst, which convincingly verify the conclusions of this study. Secondly, the authors have supplemented a plenty of details. For example, the authors have supplied the details to optimize the catalyst loadings for HER, HzOR, DHzFC and OHzS and the hydrazine concentration for the H₂ production rate of the self-powered system. In addition to the above optimizations of the four loadings and the concentration, I still remembered that the authors had given the details to optimize the Fe atomic percentage of the catalyst in the initially submitted manuscript. That is to say, many crucial parameters of this study were obtained not by arbitrary selections but by systemic optimizations. Moreover, the authors have also complemented many new experiments and DFT calculations, such as TEM, XRD, EIS, BET, ECSA, HzOR with different hydrazine concentrations, XANES, the synthesis and HzOR of FeS₂ nanosheets, DFT on FeS₂, and DHzFC with air.

The workload of these new works is equivalent to performing a new paper. More importantly, these new work and the newly added discussion about the novelty and the detoxification, together with the results already presented in the initially submitted manuscript, convincingly prove the excellence, the significance and the novelty of this study in the fields of electrocatalysts and energy. I have roughly checked through the literature and found that although CoS₂-/FeS₂-based materials and Fe doping have been reported, no Fe-doped CoS₂ nanosheets with the CoS₂ crystal structure unchanged have been published before this study. I also found that no self-powered H₂ production systems with a sole consumable have been reported before this study. Therefore, I am satisfied with the revised version of the paper. I believe that this paper is publishable in Nature Communications.

Reviewer #3:

Remarks to the Author:

The authors have thoroughly addressed concerns raised in my previous review. I now recommend its publication.

Responses to the Comments

Responses to the comments of Reviewer #1:

Comment:

The paper is greatly improved, so I suggest it to be accepted for publication in its present form.

Response:

Thank you very much for reviewing our revised manuscript. We greatly appreciate your inspiring and constructive comments.

Responses to the comments of Reviewer #2:

Comment:

I read it carefully and found that the authors have made a commendable effort to address all the concerns of mine and the other reviewers'. Firstly, the authors have fully complemented the stability confirmation, stability origins and surface information of the catalyst, which convincingly verify the conclusions of this study. Secondly, the authors have supplemented a plenty of details. For example, the authors have supplied the details to optimize the catalyst loadings for HER, HzOR, DHzFC and OHzS and the hydrazine concentration for the H₂ production rate of the self-powered system. In addition to the above optimizations of the four loadings and the concentration, I still remembered that the authors had given the details to optimize the Fe atomic percentage of the catalyst in the initially submitted manuscript. That is to say, many crucial parameters of this study were obtained not by arbitrary selections but by systemic optimizations. Moreover, the authors have also complemented many new experiments and DFT calculations, such as TEM, XRD, EIS, BET, ECSA, HzOR with different hydrazine concentrations, XANES, the synthesis and HzOR of FeS₂ nanosheets, DFT on FeS₂, and DHzFC with air.

The workload of these new works is equivalent to performing a new paper. More importantly, these new work and the newly added discussion about the novelty and the detoxification, together with the results already presented in the initially submitted manuscript, convincingly prove the excellence, the significance and the novelty of this study in the fields of electrocatalysts and energy. I have roughly checked through the literature and found that although CoS₂-/FeS₂-based materials and Fe doping have been reported, no Fe-doped CoS₂ nanosheets with the CoS₂ crystal structure unchanged have been published before this study. I also found that no self-powered H₂ production systems with a sole consumable have been reported before this study. Therefore, I am satisfied with the revised version of the paper. I believe that this paper is publishable in Nature Communications.

Response:

We are very grateful to your encouraging and positive comments and really appreciate your agreement of acceptance with this revised manuscript.

Responses to the comments of Reviewer #3:

Comment:

The authors have thoroughly addressed concerns raised in my previous review. I now recommend its publication.

Response:

We truly thank you for reviewing the revised version of our manuscript and greatly appreciate your helpful and affirmative comments.